# Astrocytes modulate brainstem respiratory rhythm-generating circuits and determine exercise capacity

Shahriar Sheikhbahaei [1,2], Egor A. Turovsky[1,3], Patrick S. Hosford[1], Anna Hadjihambi[1], Shefeeq M. Theparambil[1], Beihui Liu[4], Nephtali Marina[5], Anja G. Teschemacher[4], Sergey Kasparov[4,6], Jeffrey C. Smith[2] & Alexander V. Gourine [1]

Astrocytes are implicated in modulation of neuronal excitability and synaptic function, but it remains unknown if these glial cells can directly control activities of motor circuits to influence complex behaviors in vivo. This study focused on the vital respiratory rhythm-generating circuits of the preBötzinger complex (preBötC) and determined how compromised function of local astrocytes affects breathing in conscious experimental animals (rats). Vesicular release mechanisms in astrocytes were disrupted by virally driven expression of either the dominant-negative SNARE protein or light chain of tetanus toxin. We show that blockade of vesicular release in preBötC astrocytes reduces the resting breathing rate and frequency of periodic sighs, decreases rhythm variability, impairs respiratory responses to hypoxia and hypercapnia, and dramatically reduces the exercise capacity. These findings indicate that astrocytes modulate the activity of CNS circuits generating the respiratory rhythm, critically contribute to adaptive respiratory responses in conditions of increased metabolic demand and determine the exercise capacity.

[1] Centre for Cardiovascular and Metabolic Neuroscience, Department of Neuroscience, Physiology and Pharmacology, University College London, London, WC1E 6BT, UK. [2] Cellular and Systems Neurobiology Section, National Institute of Neurological Disorders and Stroke (NINDS), National Institutes of Health (NIH), Bethesda, MD 20892, USA. [3] Institute of Cell Biophysics, Russian Academy of Sciences, Pushchino, Russian Federation 142290. [4] Department of Physiology and Pharmacology, University of Bristol, Bristol, BS8 1TD, UK. [5] Department of Clinical Pharmacology and Experimental Therapeutics, University College London, London, WC1E 6BT, UK. [6] Baltic Federal University, Kaliningrad, Russian Federation 236041. Correspondence and requests for materials should be addressed to J.C.S. (email: smithj2@ninds.nih.gov) or to A.V.G. (email: a.gourine@ucl.ac.uk)

Astrocytes have been proposed to modulate neuronal excitability, synaptic transmission, and plasticity[1,2]. Physiology of these electrically non-excitable cells of the brain is governed by intracellular $Ca^{2+}$, with increases in $[Ca^{2+}]_i$ triggering release of signaling molecules or "gliotransmitters" (such as ATP/adenosine, D-serine, and others). Recent studies have suggested that via release of gliotransmitters astrocytes may influence activities of neural circuits controlling sleep, feeding, and chemosensing[3–5], yet it remains unknown whether astrocytes can directly modulate motor circuits and have an impact on complex behaviors. In vitro experiments with rodent brainstem slices[6–9] have suggested that astroglial mechanisms may play a certain role in regulating the activities of neuronal networks producing motor rhythms, including those within the preBötzinger complex (preBötC)[10] in the ventrolateral medulla that generates the rhythm of breathing[11]. However, whether such modulation is functionally important for rhythmic motor behavior has not been determined. In this study, we accordingly focused on the preBötC that produces a fundamental, clearly defined motor output, and where local astrocytic modulation of neuronal excitability and/or synaptic transmission would directly affect respiratory motor behavior. We determined the effects of compromised preBötC astroglial vesicular release mechanisms on breathing in conscious adult rats at rest and in conditions of increased metabolic demand requiring regulatory adjustments of respiratory motor activity, including during exercise. We show that blockade of vesicular release in preBötC astrocytes reduces the resting breathing rate and frequency of periodic sighs, decreases rhythm variability, impairs respiratory responses to hypoxia and hypercapnia, and dramatically reduces the exercise capacity.

## Results

Vesicular release mechanisms in preBötC astrocytes in adult Sprague-Dawley male rats were disrupted by virally driven expression of either the light chain of tetanus toxin (TeLC)[12], or the dominant-negative SNARE (dnSNARE) protein[13] (Supplementary Table 1) to block SNARE-dependent vesicular exocytosis. Astrocyte-specific expression of TeLC or dnSNARE was controlled by an enhanced GFAP promoter[5] (Fig. 1a). The high efficacy of TeLC expression in blocking vesicular release in brainstem astrocytes has been demonstrated previously[12]. To determine efficacy of our novel dnSNARE construct, we used total internal reflection fluorescence microscopy (TIRF) to monitor vesicular fusion events in cultured brainstem astrocytes transduced to express dnSNARE or a control transgene (CatCh-EGFP). In dnSNARE-expressing astrocytes, the number of juxtamembrane vesicles labeled with quinacrine was reduced by 67% ($p < 0.001$; Fig. 1b). Facilitated vesicular fusion induced by the $Ca^{2+}$ ionophore ionomycin, or the oxygen scavenger sodium dithionite, was effectively abolished in astrocytes expressing dnSNARE (Fig. 1c–e; Supplementary Fig. 1).

In conscious rats, bilateral expression of dnSNARE or TeLC in preBötC astrocytes (Fig. 1f; Supplementary Figs. 2 and 3) resulted in a significant reduction in resting breathing frequency ($f_R$) by 11% ($94\pm2$ vs. $106\pm5$ min$^{-1}$ in controls; $n = 5$, $p = 0.016$) and by 11% ($92\pm2$ vs. $103\pm3$ min$^{-1}$ in controls; $n = 12$, $p = 0.011$), respectively (Fig. 1g, h). Since dnSNARE or TeLC expression in astrocytes is likely to block exocytosis of several putative gliotransmitters, we determined the possible contribution of ATP by blocking ATP-mediated signaling within the preBötC by virally driven expression of a potent ectonucleotidase — transmembrane prostatic acid phosphatase (TMPAP). TMPAP expression is highly effective in preventing ATP accumulation in astroglial vesicular compartments and blocking extracellular ATP actions[14–16]. We found that bilateral expression of TMPAP in the

preBötC (Supplementary Figs. 2 and 3) reduced resting $f_R$ by 12% ($98\pm3$ vs. $111\pm4$ min$^{-1}$ in controls, $n = 7$, $p = 0.017$; Fig. 1i). These results suggested that at rest, vesicular release of gliotransmitters by preBötC astrocytes provides tonic excitatory drive to the inspiratory rhythm-generating circuits.

We next assessed whether activation of preBötC astrocytes influences breathing behavior. Release of gliotransmitters by astrocytes may occur following activation of phospholipase C (PLC)[12]. To facilitate PLC-mediated release of gliotransmitters, we transduced preBötC astrocytes to express a $G_q$-coupled Designer Receptor Exclusively Activated by Designer Drug (DREADD$_{Gq}$)[17] (see vector layout, Fig. 2a, f–h; Supplementary Figs. 2 and 3). As expected, the DREADD ligand clozapine-N-oxide (CNO) triggered robust increases in $[Ca^{2+}]_i$ in brainstem astrocytes expressing DREADD$_{Gq}$ (Fig. 2b; Supplementary Fig. 4). These responses were blocked by the PLC inhibitor U73122 (Fig. 2c). However, a PLC activity assay revealed higher resting (i.e., in the absence of CNO) levels of inositol phosphates in cultured astrocytes expressing DREADD$_{Gq}$ (Fig. 2d). Moreover, DREADD$_{Gq}$ expression was also found to be associated with a higher rate of spontaneous fusion of quinacrine-labeled vesicles in cultured astrocytes (Fig. 2e), and facilitated release of ATP in conditions when preBötC astrocytes were transduced to express the transgene (in experiments on acute brainstem slices, Fig. 2i, j), indicating that in the absence of an agonist, DREADD$_{Gq}$ is constitutively active at the level of expression achieved by the viral vector used. We exploited this property of DREADD$_{Gq}$ in order to determine whether sustained activation of PLC in preBötC astrocytes, associated with facilitated vesicular release of ATP, has an impact on the inspiratory rhythm-generating circuits. Bilateral expression of DREADD$_{Gq}$ ($n = 8$) in preBötC astrocytes resulted in 26% higher baseline $f_R$ ($123\pm5$ vs. $98\pm2$ min$^{-1}$ in controls, $n = 14$, $p < 0.001$; Fig. 2k). This effect was effectively abolished by the ectonucleotidase activity of TMPAP. Co-expression of DREADD$_{Gq}$ and TMPAP in the preBötC was associated with a significant reduction of the respiratory rate below the baseline ($88\pm3$ min$^{-1}$, $n = 5$, $p = 0.030$; Fig. 2k), an effect similar to that observed in conditions of TMPAP expression alone (Fig. 1i).

Altered function of preBötC astrocytes also had a significant impact on other features of resting inspiratory activity. Bilateral expression of dnSNARE or TeLC in preBötC astrocytes was associated with a significant reduction in the variability of the respiratory rhythm (Fig. 3a). DREADD$_{Gq}$ expression had an opposite effect and increased respiratory variability (Fig. 3a).

The frequency of sighs, breaths with augmented inspiration, generated periodically by the preBötC circuits[18,19], was reduced by 27% ($p < 0.001$) in rats expressing dnSNARE, by 25% ($p < 0.001$) in rats expressing TeLC, and by 26% ($p < 0.001$) in rats expressing TMPAP in the preBötC (Fig. 3b). Sigh frequency was found to be significantly higher in rats transduced to express DREADD$_{Gq}$ in preBötC astrocytes (by 31%, $n = 8$, $p < 0.001$; Fig. 3b). There is recent evidence that generation of sighs is facilitated by the actions of bombesin-like peptides[18] and inhibited when astroglial function is compromised[20], suggesting that sigh generation may be modulated by signaling molecules released by preBötC astrocytes in response to various stimuli, including locally released bombesin-like peptides. Indeed, we found that bombesin triggers robust $[Ca^{2+}]_i$ responses in cultured brainstem astrocytes (Supplementary Fig. 5). Blockade of vesicular release mechanisms in preBötC astrocytes (dnSNARE expression) significantly reduced the effect of bombesin on sigh frequency in vivo (Supplementary Fig. 6), suggesting that the actions of bombesin-like peptides on preBötC circuits[18] are potentially mediated by astrocytes. Together these results suggest that vesicular release of gliotransmitter(s) by preBötC astrocytes

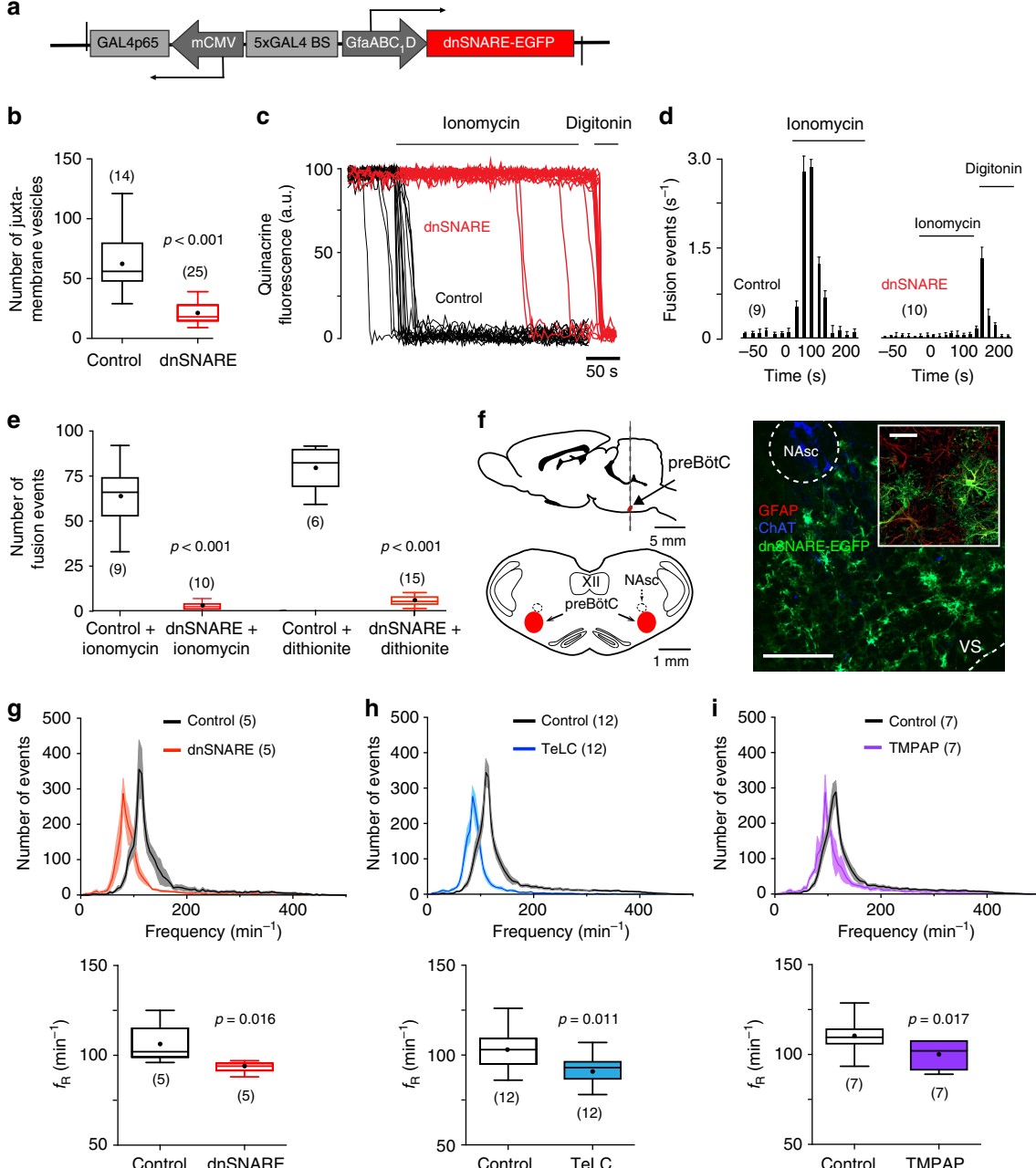

**Fig. 1** PreBötC astrocytes modulate the activity of the respiratory rhythm-generating circuits. **a** Schematic of AVV-sGFAP-dnSNARE-EGFP vector layout. **b** Summary data illustrating a reduction in the number of juxtamembrane quinacrine-labeled vesicular compartments in cultured brainstem astrocytes expressing dnSNARE. **c** Plots of TIRF intensity changes showing loss of quinacrine fluorescence from a proportion of labeled organelles in response to application of the $Ca^{2+}$ ionophore ionomycin (1 µM) in two individual cultured astrocytes transduced to express control transgene (black traces) or dnSNARE (red traces). In cultures of astrocytes expressing dnSNARE, digitonin was applied at the end of the recording to permeabilize the membranes, resulting in a rapid loss of quinacrine fluorescence. **d** Averaged temporal profile of ionomycin-induced vesicular fusion events detected in cultured astrocytes expressing control transgene or dnSNARE. **e** Total number of ionomycin- and sodium dithionite-induced vesicular fusion events detected in individual cultured astrocytes expressing control transgene or dnSNARE. In **b**, **d**, and **e**, numbers of individual tests performed in three different cultures prepared from different animals are indicated. **f** Schematic drawings of the rat brain in parasagittal and coronal projections illustrating the location of the preBötC. NAsc, semi-compact division of the nucleus ambiguus; XII, hypoglossal motor nucleus. Representative confocal image of dnSNARE-EGFP expression in preBötC astrocytes is shown on the right (scale bar: 200 µm). High-magnification inset shows expression of dnSNARE-EGFP in GFAP-positive preBötC astrocytes (inset scale bar: 50 µm). NAsc neurons are identified by choline acetyltransferase (ChAT) immunoreactivity. VS, ventral surface of the brainstem. **g**, **h** Group data showing the effects of dnSNARE or TeLC expression in preBötC astrocytes on frequency distribution of all respiratory-related events detected in 30-min assay (top) and resting respiratory frequency recorded during periods of calm wakefulness and/or quiet sleep ($f_R$, bottom) in conscious adult rats. In control animals preBötC astrocytes were transduced to express CatCh-EGFP. **i** Group data showing the effect of TMPAP expression in preBötC region on frequency distribution of the respiratory-related events and resting $f_R$ in conscious rats. In **g**, **h**, and **i**, number of animals in each experimental group is indicated in parentheses. p values—Mann–Whitney U rank test

modulates the variability of the respiratory rhythm and the generation of sighs.

Since hypoxia induces release of ATP by astrocytes[12,21] and increases sigh frequency[22,23], we next evaluated the effects of dnSNARE or TeLC expression in preBötC astrocytes on respiratory responses to systemic hypoxia (10% $O_2$ in the inspired air) as well as the effects of dnSNARE, TeLC, TMPAP, and $DREADD_{Gq}$ expression on sigh generation during hypoxia. Expression of dnSNARE attenuated hypoxia-induced increases in $f_R$ by 27% (159±10 vs. 217±7 min$^{-1}$ in controls; Fig. 4a) and in

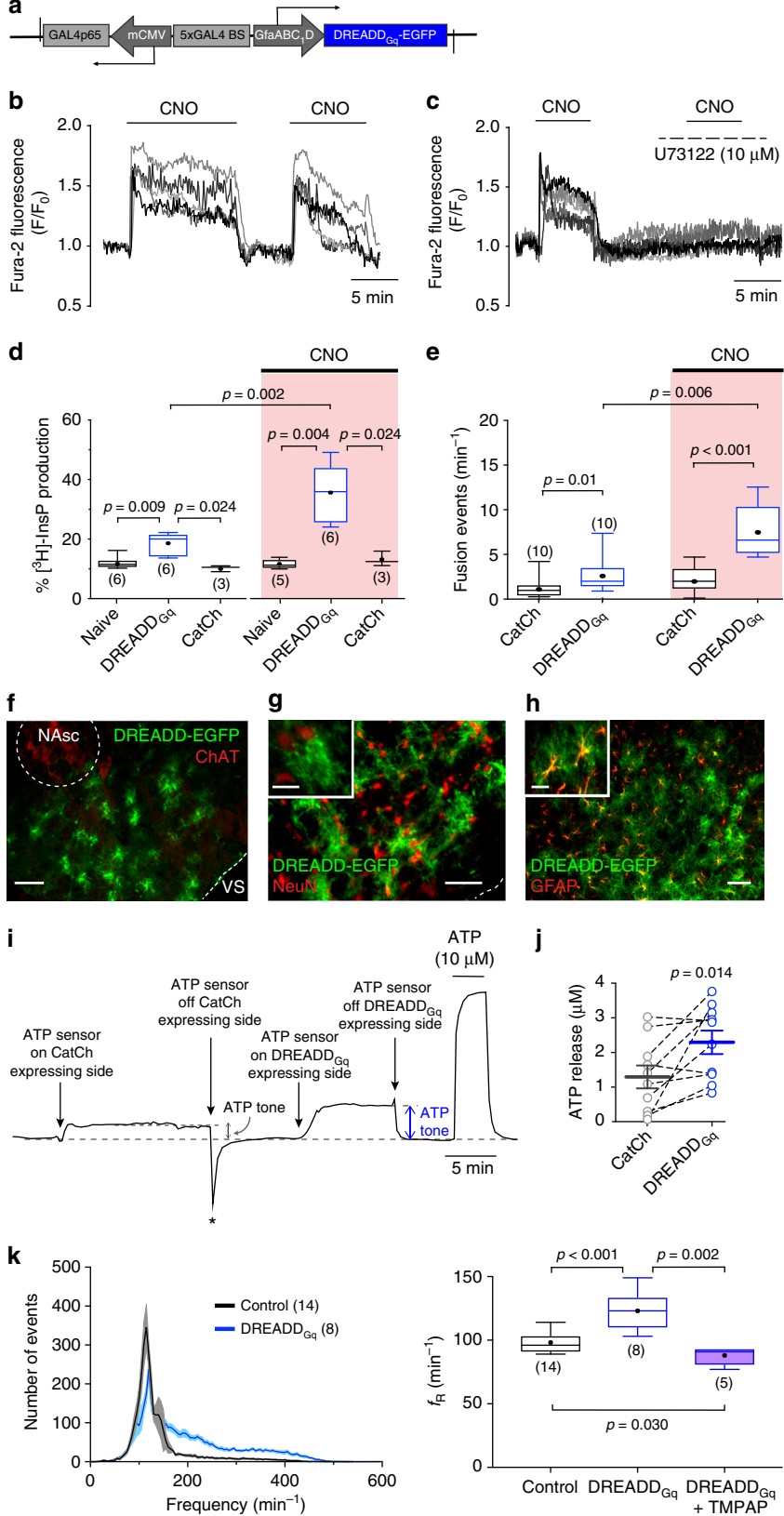

minute ventilation by 34% (Fig. 4a). TeLC expression in preBötC astrocytes had a similar effect (Supplementary Fig. 6). Disruption of either astroglial vesicular release (dnSNARE or TeLC expression) or ATP-mediated signaling (TMPAP expression) reduced the frequency of sighs during the hypoxic challenge by 34% ($n = 5$, $p < 0.001$), 36% ($n = 12$, $p < 0.001$), and 44% ($n = 7$, $p < 0.001$), respectively (Fig. 3b). DREADD$_{Gq}$ expression in preBötC astrocytes had an opposite effect and increased frequency of sigh generation during hypoxia by 50% ($n = 8$, $p = 0.003$; Fig. 3b).

Brainstem astrocytes are sensitive to changes in $PCO_2$/pH[24–26] and we next found that preBötC astrocytes play an important role in the development of respiratory response to elevated systemic $CO_2$ (hypercapnia). In conscious rats, bilateral expression of dnSNARE or TeLC in preBötC astrocytes reduced the $f_R$ responses to hypercapnia (6% inspired $CO_2$) by 23% (141±6 vs. 182±3 min$^{-1}$ in controls; $n = 5$, $p = 0.008$) and 20% (151±6 vs. 190±8 min$^{-1}$ in controls; $n = 9$, $p = 0.005$), respectively (Fig. 4b), concomitantly reducing minute ventilation (Fig. 4b).

We next hypothesized that astroglial control of breathing at the level of the preBötC may become particularly important during physical activity and exercise when increased oxygen demand must be supported by an enhanced respiratory effort. Accordingly, we determined whether blockade of astroglial vesicular release mechanisms impairs the exercise capacity. Bilateral expression of dnSNARE or TeLC in preBötC astrocytes resulted in a marked reduction of exercise capacity by 57% (0.5±0.1 vs. 1.1 ±0.2 kJ in controls; $p = 0.016$, $n = 5$) and 42% (0.7±0.1 vs. 1.2±0.1 kJ in controls; $p < 0.001$, $n = 9$), respectively (Fig. 5a, b). Cardiovascular responses to exercise (increases in heart rate and systemic arterial blood pressure) were not affected (Fig. 5c), suggesting that the impaired exercise capacity is due to the respiratory, not a cardiovascular, deficit.

## Discussion

Central nervous system (CNS) neural circuits are intermingled with astroglial networks, yet the experimental evidence for astrocytes directly controlling the activities of functionally defined motor circuits to affect behavior in vivo is lacking. This study focused on the vital brainstem respiratory rhythm-generating circuits of the preBötC and determined whether astrocytes can modulate the activity of these circuits and, therefore, breathing in conscious rats. Astrocytes are well known to provide neurons with structural and metabolic support, but also have a distinct signaling function, which is mediated via the release of gliotransmitters[2]. Here molecular approaches designed to block or stimulate astroglial vesicular release of gliotransmitters were used to study the functional role of astroglial signaling in the control of the respiratory activity originating at the level of the preBötC.

Expression of TeLC or dnSNARE protein in preBötC astrocytes reduced the resting breathing rate and frequency of periodic sighs, decreased rhythm variability, impaired respiratory responses to hypoxia and hypercapnia, and dramatically reduced the exercise capacity. TeLC is a proteolytic enzyme that cleaves SNARE proteins required for vesicular fusion. In cultured astrocytes, TeLC inhibits ATP and glutamate release[27], blocks $Ca^{2+}$-dependent vesicular fusion[12], and prevents the spread of $Ca^{2+}$ waves triggered by mechanical stimulation[12]—the effects consistent with the inhibition of vesicular ATP release[28]. TIRF imaging confirmed that dnSNARE expressed in astrocytes reduces the number of juxtamembrane vesicles and effectively blocks $Ca^{2+}$-dependent vesicular fusion. These effects are in line with the proposed mechanisms underlying the effect of dnSNARE on exocytosis in astrocytes[29].

Our initial design of the gain-of-function experiment with DREADD$_{Gq}$ involved targeting preBötC astrocytes to express this receptor followed by documenting changes in respiratory activity induced by administration of CNO. However, validation experiments of our viral vector construct revealed that in the absence of a ligand, astrocytes expressing DREADD$_{Gq}$ exhibit a higher level of PLC activity, higher rate of spontaneous vesicular fusion, and facilitated tonic release of ATP. These results suggested that DREADD$_{Gq}$ is constitutively active when expressed in astrocytes, an observation consistent with the properties of many hM3 receptor mutants originally described[17]. Since CNO appears to have low affinity for DREADDs and its effects are largely attributed to its conversion to clozapine[30], which may interact with astroglial serotonin receptors[31], we focused on determining the effects of the constitutive DREADD$_{Gq}$ activity. In rats, sustained activation of PLC-mediated signaling pathways in preBötC astrocytes expressing DREADD$_{Gq}$ was associated with higher resting breathing rate, higher frequency of periodic sighs, and increased rhythm variability. That this effect was blocked by co-expression of the potent ectonucleotidase TMPAP suggested that the stimulatory effect of DREADD$_{Gq}$ expression in preBötC astrocytes on breathing could be mediated by direct actions of ATP and/or related purines on preBötC circuits or, alternatively, autocrine effects of ATP on $Ca^{2+}$ in preBötC astrocytes leading to the release of other gliotransmitters[2], neither possibility of which we rule out here.

The role of preBötC astrocytes in the control of breathing becomes especially important during physiological metabolic challenges, such as systemic hypoxia and hypercapnia, where enhanced respiratory effort is critical to maintain homeostasis. Although expression of dnSNARE or TeLC in preBötC astrocytes reduced resting respiratory rate, minute ventilation at normoxia/eucapnia was similar to that in animals expressing control

**Fig. 2** Activation of G$_q$-mediated signaling pathways in preBötC astrocytes facilitates the respiratory rhythm. **a** Schematic of AVV-sGFAP-DREADD$_{Gq}$-EGFP vector layout. **b** CNO-induced [$Ca^{2+}$]$_i$ responses in cultured astrocytes transduced to express DREADD$_{Gq}$. **c** The effect of CNO is blocked by the PLC inhibitor U73122. **d** Summary data illustrating PLC activity in cultured brainstem astrocytes expressing DREADD$_{Gq}$ and the effects of CNO on PLC activity as assessed by measuring [$^3$H]-inositol phosphate (InsP) production relative to the total inositol lipid pool in naive astrocytes and in astrocytes transduced to express DREADD$_{Gq}$ or CatCh. Higher resting InsP level in brainstem astrocytes expressing DREADD$_{Gq}$ indicate constitutive activity of the receptor. **e** Summary data illustrating the rate of spontaneous and CNO-induced fusion of quinacrine-labeled vesicular compartments in astrocytes transduced to express CatCh or DREADD$_{Gq}$. DREADD$_{Gq}$ expression in astrocytes is associated with a significantly higher rate of spontaneous vesicular fusion events. In **d** and **e**, numbers of individual tests performed in three different cultures prepared from different animals are indicated. **f–h** Confocal images illustrating DREADD$_{Gq}$-EGFP expression in a subset of preBötC astrocytes (scale bars: 100 μm). DREADD$_{Gq}$ expression is limited to astrocytes as no neurons (identified by NeuN immunoreactivity) expressed the transgene (see high-magnification inset images. Inset scale bars: 25 μm). **i** Representative example of changes in ATP biosensor current after biosensor placement on the surface of the brainstem slice transduced to express CatCh and DREADD$_{Gq}$ in astrocytes residing in opposite preBötC areas. Asterisk (*) marks the movement deflection due to removing of the biosensor from the slice surface. **j** Summary data illustrating facilitated tonic release of ATP in acute brainstem slices of adult rats transduced to express DREADD$_{Gq}$ by the preBötC astrocytes; **k** Group data showing the effect of DREADD$_{Gq}$ expression in preBötC astrocytes on frequency distribution of all respiratory-related events and resting $f_R$ in conscious rats. Number of animals in each experimental group is indicated in parentheses. $p$ values—Mann–Whitney $U$ rank test (**d**, **e**, **k**) or Wilcoxon matched-pairs signed-rank test (**j**)

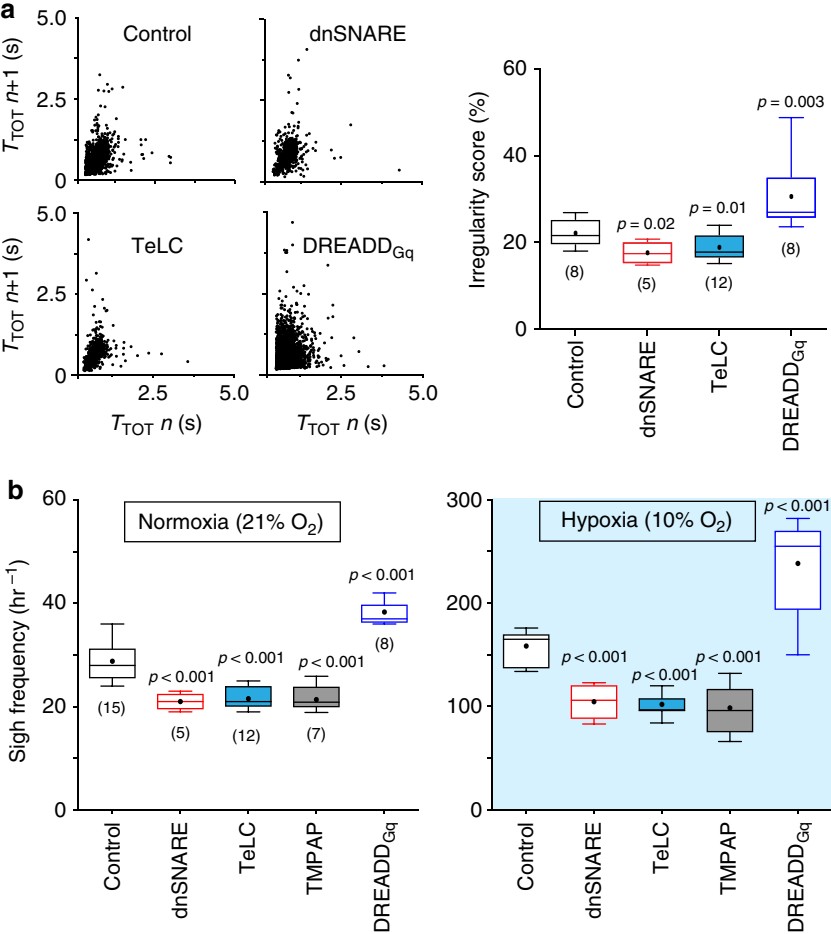

**Fig. 3** PreBötC astrocytes modulate the variability of the respiratory rhythm and the generation of sighs. **a** Regularity of the respiratory rhythm in conditions of activation or blockade of vesicular release mechanisms in preBötC astrocytes. Poincaré plots of the respiratory cycle duration ($T_{TOT}$) for the $n$th cycle vs. $T_{TOT}$ for the $n$th + 1 cycle in rats transduced to express the control transgene (CatCh), dnSNARE, TeLC, or DREADD$_{Gq}$ by the preBötC astrocytes. Right: summary data illustrating irregularity score of the respiratory rhythm in conscious rats transduced to express CatCh, dnSNARE, TeLC, or DREADD$_{Gq}$. **b** Summary data illustrating sigh frequency at resting conditions and during systemic hypoxia (10% $O_2$ in the inspired air) in conscious rats transduced to express control transgenes, dnSNARE, TeLC, TMPAP, or DREADD$_{Gq}$ by preBötC astrocytes. Number of animals in each experimental group is indicated in parentheses. $p$ values—Mann–Whitney $U$ rank test

transgene (due to small compensatory increases in tidal volume). More marked differences in ventilation between the experimental and control groups were observed during the hypoxic challenge, indicating that preBötC astrocytes are critically important for the development of the full-scale hypoxic ventilatory response. These data are consistent with the proposed role of astrocytes as CNS oxygen sensors[12,21,32].

Respiratory rhythm-generating circuits are silent in the absence of $CO_2$ and require a certain level of $CO_2$ to operate. The preBötC has a neuronal $H^+/CO_2$-sensing mechanism[33], however, our results suggest that preBötC astrocytes contribute in a significant manner to the development of the respiratory response to hypercapnia. Our data support the "distributed central chemosensitivity" hypothesis, which proposes that central respiratory sensitivity to $CO_2$ (the mechanism that adjusts breathing in accordance with changes in brain parenchymal $PCO_2/pH$) is mediated by multiple central chemoreceptor sites (one being the preBötC), with each site providing tonic excitation in eucapnia and a fraction of the total response to systemic hypercapnia[34]. Previous experimental studies suggested that the contribution of the preBötC mechanism(s) to the overall respiratory response to $CO_2$ is ~20–25%[34]. Our experiments showed that ventilation during eucapnia and hypercapnia is similarly reduced by ~20% in

conditions when vesicular release mechanisms in preBötC astrocytes are blocked and hyperoxia is applied to reduce the drive from the peripheral chemoreceptors (Fig. 4b). While current models of central respiratory $CO_2$ chemosensitivity are focused on groups of pH-sensitive neurons residing elsewhere in the brainstem[35–37], our data suggest that $CO_2/pH$ chemosensitivity of the preBötC is mediated by astrocytes.

Recently, astrocytes have been implicated in the brain mechanisms that maintain endurance capacity[38], although the specific contribution of brainstem and especially preBötC astroglia and the potential underlying mechanisms of this involvement were not addressed. We found that expression of dnSNARE or TeLC in preBötC astrocytes is associated with a significant reduction in exercise capacity. The exact mechanism underlying the involvement of preBötC astrocytes in the development of the respiratory response to exercise remain to be determined, but these data imply that astrocytes intermingled with the preBötC respiratory neural circuits ultimately determine the exercise capacity.

In conclusion, the data obtained in the present study indicate that astrocytes are able to modulate the activities of vital rhythmic motor circuits with a significant impact on motor behavior in vivo. We targeted astrocytes intermingled with the preBötC

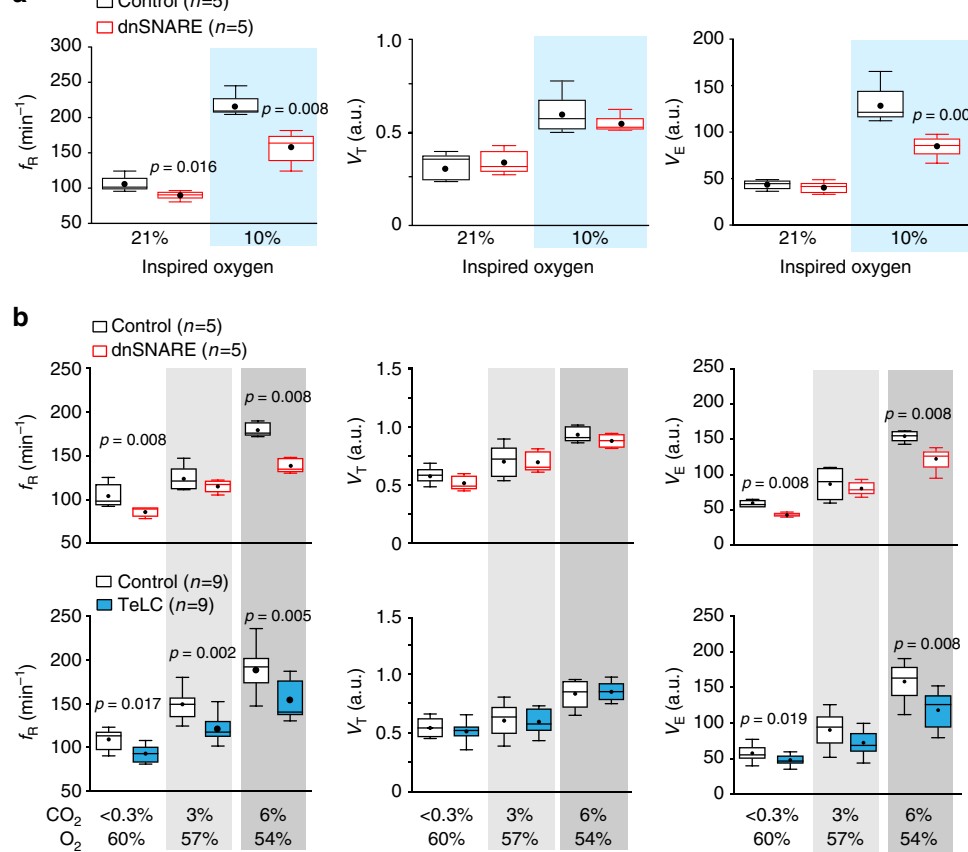

**Fig. 4** Astrocyte signaling within the preBötC contributes to the development of the respiratory responses to systemic hypoxia and hypercapnia. **a** Summary data illustrating the effect of dnSNARE expression in preBötC astrocytes on hypoxia-induced increases in the respiratory frequency ($f_R$), tidal volume ($V_T$), and minute ventilation ($V_E$) in conscious rats. **b** Group data illustrating the effect of dnSNARE or TeLC expression in preBötC astrocytes on $CO_2$-induced increases in $f_R$, $V_T$, and $V_E$ in conscious rats. The significant reduction in $V_E$ during systemic hypoxia or hypercapnia in rats transduced to express dnSNARE or TeLC in preBötC astrocytes is attributable to the smaller increases in $f_R$ without significant effects of either transgene on $V_T$. Number of animals in each experimental group is indicated in parentheses. $p$ values—Mann–Whitney $U$ rank test. Data sets without $p$ values indicated are not significantly different

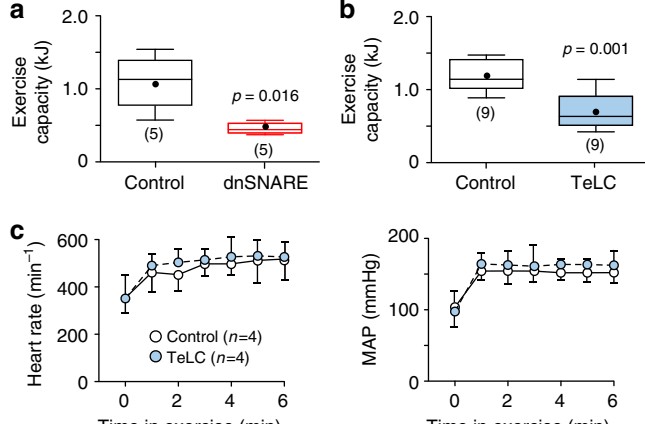

**Fig. 5** PreBötC astrocytes determine the exercise capacity. **a**, **b** Summary data illustrating the effects of dnSNARE or TeLC expression in preBötC astrocytes on exercise capacity. $p$ values—Mann–Whitney $U$ rank test. **c** TeLC expression in preBötC astrocytes had no effect on the cardiovascular responses to exercise. MAP—mean arterial blood pressure. Number of animals in each experimental group is indicated in parentheses. Data are presented as means ± SEM

respiratory rhythm-generating circuits to express proteins that block or facilitate vesicular release mechanisms. Our results suggest that astroglial signaling involving exocytotic vesicular release of gliotransmitters provides tonic excitation of preBötC circuits that generate the inspiratory rhythm. The role of preBötC astrocytes becomes especially important in conditions such as systemic hypoxia, hypercapnia, and exercise, when homeostatic adjustments of breathing are critical to support our physiological and behavioral demands.

## Methods

**Animals**. All animal experiments were performed in Sprague-Dawley rats (adult males 250–270 g or neonates P2-P3 of either sex) in accordance with the European Commission Directive 2010/63/EU (European Convention for the Protection of Vertebrate Animals used for Experimental and Other Scientific Purposes), the UK Home Office (Scientific Procedures) Act (1986), and the National Institutes of Health Guide for the Care and Use of Laboratory Animals, with project approval from the respective Institutional Animal Care and Use Committees. Animals were housed in a temperature-controlled facility with a normal light-dark cycle (12 h:12 h, lights on at 0700 hours). Tap water and laboratory rodent chow were provided ad libitum.

**Molecular approaches to block astroglial signaling**. To block vesicular release mechanisms in preBötC astrocytes, we developed a novel adenoviral vector (AVV) to drive the expression of dnSNARE protein[13] under the control of an enhanced GFAP promoter (Fig. 1a; Supplementary Table 1). Description of validation of the transgene efficacy in blocking vesicular release mechanisms in astrocytes is

provided above. PreBötC astrocytes were also targeted to express TeLC, which blocks vesicular exocytosis via proteolytic degradation of SNARE proteins. Generation of the AVV to drive the expression of TeLC (Supplementary Table 1; AVV-sGFAP-EGFP-skip-TeLC) in astrocytes and validation of the transgene efficacy in blocking vesicular release and signaling between astrocytes were described in detail previously[12]. To block ATP-mediated signaling we used a lentiviral vector (LVV) to overexpress a potent ectonucleotidase TMPAP[39]. Expression of TMPAP was driven under the control of an elongation factor 1α (EF1α) promoter (Supplementary Table 1; LVV-Ef1α-TMPAP-EGFP). High efficacy of TMPAP enzymatic activity in preventing vesicular ATP accumulation in astrocytes and blocking ATP-mediated signaling between astrocytes was characterized in detail previously[14,15].

**Molecular approach to activate astrocytes.** To stimulate $G_q$-coupled signaling pathways we generated an AVV to express $DREADD_{Gq}$ fused with enhanced green fluorescent protein (EGFP) in astrocytes (Fig. 2a; Supplementary Table 1). $DREADD_{Gq}$ expressed in astrocytes was found to be constitutively active. This constitutive activity of $DREADD_{Gq}$ was lower compared to that triggered by the application of the DREADD ligand CNO (Fig. 2d, e) and did not appear to be detrimental to the cells expressing the transgene. Brainstem astrocytes transduced to express $DREADD_{Gq}$ looked normal upon histological examination (Fig. 2f–h) and were able to mount unaltered $[Ca^{2+}]_i$ responses to activation of purinoceptors following application of ATP (Supplementary Fig. 4).

**Control transgenes.** Two control vectors were used: (1) an AVV to express calcium translocating channelrhodopsin variant (CatCh) fused with EGFP (CatCh-EGFP) under the control of the GFAP promoter (AVV-sGFAP-CatCh-EGFP); and (2) a LVV to express EGFP under the control of EF1α promoter (despite the use of the generic promoter, EGFP expression driven by LVVs is almost exclusively confined to astroglia[14]) (Supplementary Table 1). The choice of CatCh-EGFP was dictated by the need of having a transduced membrane-bound protein as an appropriate control in the experiments where brainstem astrocytes were transduced to express $DREADD_{Gq}$, which is also a membrane protein. Parts of both CatCh and DREADDs are facing the extracellular space and (as foreign proteins) may potentially trigger an immune response. Thus, expression of CatCh-EGFP is a much harsher control than cytoplasmic expression of EGFP, which has hardly ever been reported to cause any adverse cellular effects. Each experimental (dnSNARE, TeLC, $DREADD_{Gq}$, or TMPAP) and control (CatCh-EGFP or EGFP) animal groups were injected with the appropriate viral vector at the same time and the experimental groups were compared to their own control groups (Figs. 1g–i and 2k). These experiments were conducted over the course of 24 months and some variations in the baseline respiratory frequency were observed (Supplementary Fig. 3f). However, these differences in resting respiratory activity between animals from different control groups expressing CatCh-EGFP or EGFP in the preBötC and naive (non-transduced) rats were not statistically significant (Supplementary Fig. 3f). Since no significant variations across all the control groups were observed when the frequency of respiratory sighs was analyzed, the sigh frequency data obtained in eight representative control animals transduced to express CatCh-EGFP and seven control animals transduced to express EGFP were combined and used for the analysis and data presentation (Fig. 3b).

**In vivo viral gene transfer.** Adult male rats (250–270 g) were anesthetized with a mixture of ketamine (60 mg kg$^{-1}$, intramuscular (i.m.)) and medetomidine (250 µg kg$^{-1}$, i.m.) and placed in a stereotaxic frame. The tooth bar was adjusted so that bregma was positioned 5 mm below lambda. PreBötC areas were targeted bilaterally by advancing a pipette from the dorsal surface of the brainstem. Viral vectors (see Supplementary Table 1 for viral titers) were delivered via a single microinjection (0.20–0.25 µl) per side using the following coordinates: 0.9 mm rostral; 2 mm lateral; and 2.7 mm ventral from the *calamus scriptorium*. After the microinjections, the wound was sutured and anesthesia was reversed with atipamezole (1 mg kg$^{-1}$, i.m.). For postoperative analgesia, the animals received buprenorphine (0.05 mg kg$^{-1}$ kg$^{-1}$ per day, subcutaneous) for 3 days. No complications were observed after the surgery and the animals gained weight normally.

**Cell culture.** Primary astroglial cell cultures were prepared from the brainstem tissue of rat pups (P2-P3) as described[12,40]. The animals were euthanized by isoflurane overdose, the brains were removed, and the ventral regions of the medulla oblongata were dissected out. Ventral brainstem tissue cuts from two to three animals were used for each cell culture preparation. After isolation, the cells were plated on poly-d-lysine-coated glass coverslips and maintained at 37 °C in a humidified atmosphere of 5% $CO_2$ and 95% air. Viral vectors to drive the expression of dnSNARE, $DREADD_{Gq}$, CatCh-EGFP or EGFP were added to the incubation medium at the time of cell culture preparation at $5 \times 10^8$–$5 \times 10^{10}$ transducing units per ml. Experiments were performed after 7–10 days of incubation.

**TIRF microscopy.** In cultured brainstem astrocytes, ATP-containing vesicular compartments were visualized by quinacrine staining (5 µM, 15 min incubation at 37 °C). The acridine derivative quinacrine is a weak base that binds ATP with high affinity and can be used to identify putative ATP-containing vesicles in living cells,

including astrocytes[12,27,40]. An Olympus TIRF microscope was used to monitor fusion events[12,40]. Fluorescence was excited at 488 nm and collected at 500–530 nm. The imaging setup included a high-numerical-aperture (NA) oil-immersion objective (×60, 1.65 NA), an inverted microscope (IX71; Olympus), and a cooled charge-coupled-device camera (Hamamatsu). Images were analyzed using Olympus Cell^tool software (Olympus). The experiments were performed at 37 °C.

**$Ca^{2+}$ imaging.** The $[Ca^{2+}]_i$ responses in individual cultured astrocytes were visualized by recording changes in fluorescence of conventional $Ca^{2+}$ indicators Fura-2 (Molecular Probes), Fluo-4, or Rhod-2 (Thermo Fisher)[5,12,24]. Cells were loaded with Fura-2 (5 µM; 40 min incubation; 37 °C), Fluo-4 (10 µM; 40 min incubation; 37 °C), or Rhod-2 (10 µM; 40 min incubation; 37 °C) with the addition of pluronic F-127 (0.005%). After the incubation with the dye, cultures were washed three times prior to the experiment. Changes in $[Ca^{2+}]_i$ were monitored by an inverted Olympus microscope with ×20 oil-immersion objective. Excitation light provided by a Xenon arc lamp was passed sequentially through a monochromator at 340, 380, and 490 nm (Cairn Research); emitted fluorescence was measured at 515 nm (Fura-2) or 565 nm (Rhod-2). All the experiments were performed at 37 °C.

**PLC activity assay.** Cultured naive astrocytes and astrocytes transduced to express $DREADD_{Gq}$ or CatCh were incubated for 18 h in M199 medium containing 10% dialyzed fetal calf serum and 1 µCi ml$^{-1}$ of [$^3$H]-inositol (specific activity 18.5 Ci mmol$^{-1}$) (37 °C; 5% $CO_2$, 95% $O_2$). Immediately prior to the assay, the incubation medium was replaced with Hanks' balanced salt solution buffer. Lithium chloride was then added to reach a final concentration of 10 mM and cultures were incubated at 37 °C for an additional 30 min. To activate $DREADD_{Gq}$, CNO (5 µM, Tocris Bioscience) was added for 20 min. Reactions were terminated by removal of the medium and the addition of 500 µl of ice-cold methanol. [$^3$H]-Inositol phosphate ([$^3$H]-InsP) production was determined by adding the samples to 2 ml Dowex columns pre-washed with a mixture of ammonium formate (2 M) and 0.1 M formic acid. Double-distilled water and a mixture of sodium tetraborate (5 mM) and sodium formate (60 mM) were used to elute unbound [$^3$H]-inositol and glycosylphosphatidylinositol, respectively. Then, a mixture of ammonium formate (1 M) and formic acid (0.1 M) was added to the column to elute total [$^3$H]-InsP into scintillation vials. The 500 µl aliquots of the eluted samples were then transferred in duplicates to liquid scintillation vials. Concentrations of [$^3$H] in [$^3$H]-InsP and total [$^3$H]-inositol lipids were detected using a Beckman LS 5801 scintillation counter (4 min, [$^3$H] DPM program). The results are presented as percentages of radioactive InsP ([$^3$H]-InsP) in the total inositol lipid pool (Fig. 2d).

**Measurements of ATP release in acute brainstem slices.** Adult rats were transduced to express $DREADD_{Gq}$ and CatCh in astrocytes of the left and right preBötC regions. After 7 days following microinjections of viral vectors, the animals were humanely killed by isoflurane overdose and the brainstem was quickly removed and placed in chilled (4–6 °C) artificial cerebrospinal fluid (aCSF; 124 mM NaCl, 3 mM KCl, 2 mM $CaCl_2$, 26 mM $NaHCO_3$, 1.25 mM $NaH_2PO_4$, 1 mM $MgSO_4$, 10 mM D-glucose saturated with 95% $O_2$/5% $CO_2$, pH 7.4) with an additional 9 mM $Mg^{2+}$. The medulla was isolated and a horizontal 400 µm-thick slice was cut parallel to the ventral medullary surface[41,42]. Recordings were made in a flow chamber (3 ml min$^{-1}$) at ~35 °C from the slices placed on an elevated grid to permit access of aCSF from both sides of the slice.

The design and operation of the ATP biosensors (Sarissa Biomedical) were described in detail previously[41]. To control for the release of nonspecific electroactive interferents, a dual recording configuration of the ATP biosensor and control (null) biosensor was used, as described[41–43]. A "null" biosensor (lacking enzymes but otherwise identical) current was subtracted from the current recorded by the ATP biosensor to give "net-ATP" readings, reporting release of ATP (Fig. 2i). Both sensors were initially placed in the recording chamber having no contact with the brainstem slice. Once a steady-state recording was achieved, the sensors were laid flat bilaterally (ATP sensor was placed randomly on either left or right side of the slice) in direct contact with the ventral surface of the slice in equivalent positions overlaying the preBötC. The sensors were left in place to achieve stable recordings of the ATP tone and then carefully lifted from the surface of the slice to allow measurement of tonic ATP release (Fig. 2i). Without removing the sensors from the recording chamber, their positions on the left (expressing $DREADD_{Gq}$) and right (expressing CatCh) sides of the brainstem slice were swapped to determine tonic ATP release from the opposite site (Fig. 2i). Sensors were calibrated before and after every recording by application of ATP (10 µM) (Fig. 2i). To convert changes in the biosensor current to changes in ATP concentration, an average of sensor calibrations before and after the recording was used.

**Measurements of respiratory activity in conscious rats.** Whole-body plethysmography was used to record respiratory activity in unrestrained conscious adult rats[44–46]. Briefly, 5–7 days after the injections of viral vectors the rats were placed in a Plexiglas recording chamber (1 l) that was flushed continuously with humidified air (21% $O_2$, 79% $N_2$; temperature 22–24 °C), at a rate of 1.2 l min$^{-1}$. In order to take into the account circadian variations of the physiological parameters, respiratory activity in all the animals was assessed at the same time of the day

(between 1100 and 1500 hours). The animals were allowed to acclimatize to the chamber environment for ~60 min followed by 30 min recording period of resting respiratory activity. For the experiments involving hypoxic challenges, the $O_2$ concentration in the inspired air was reduced to 10% (balanced with $N_2$) for 10 min. In a separate series of experiments, hypercapnia was induced by stepwise increases in $CO_2$ concentration in the respiratory gas mixture to 3% and 6% in hyperoxic environment (>50% $O_2$, balanced with $N_2$) to reduce the drive from the peripheral chemoreceptors. Each $CO_2$ concentration was maintained for 5 min. Concentrations of $O_2$ and $CO_2$ in the plethysmography chamber were monitored online using a fast-response $O_2$/$CO_2$ analyzer (ML206, AD Instruments). Data were acquired using Power1401 interface and analyzed offline using *Spike2* software (CED).

**Measurements of central respiratory drive in anesthetized rats**. Adult rats were transduced to express dnSNARE and CatCh by astrocytes of the left and right preBötC regions, respectively. After 5–7 days following microinjections of viral vectors, the animals were anesthetized (urethane, 1.5 g kg$^{-1}$, intraperitoneal (i.p.)) and instrumented for blood pressure recording via femoral artery cannulation. The depth of anesthesia was monitored by the stability of blood pressure and heart rate. The trachea was cannulated low in the neck with a 12-gauge cannula to maintain airway patency. Animals were then transferred to a stereotaxic frame and left to breathe spontaneously. An occipital craniotomy was performed to expose the dorsal aspect of the brainstem. Phrenic nerve activity (PNA) was recorded as an indicator of central inspiratory drive. The PNA signal was amplified (20 000×), filtered (500–1500 Hz), rectified, and smoothed ($\tau = 50$ ms). Arterial blood gases and pH were monitored regularly using a blood gas analyzer (Model 380, Siemens) and maintained with the physiological ranges. Core body temperature was maintained at 37.0 ± 0.5 °C. Bombesin (250 µM; 50 nl) was microinjected over a period of 10–15 s into the preBötC region using a single barreled micropipette (tip diameter 10–15 µm). The PNA response to bombesin was followed until returning to baseline (usually within 30 min) before a second microinjection was made into the contralateral preBötC. Data were recorded using Power1401 interface and analyzed offline using *Spike2* software. At the end of the experiments, the animals were humanely killed by an overdose of pentobarbitone sodium (200 mg kg$^{-1}$, intravenous).

**Biotelemetry transmitter implantation**. Systemic arterial blood pressure and heart rate in exercising animals were recorded using biotelemtry. The rats were anesthetized with ketamine (60 mg kg$^{-1}$, i.m.) and medetomidine (250 µg kg$^{-1}$, i.m.), a laparotomy was performed, and a telemetry pressure probe (model TA11PA-C40, DSI) was implanted into the abdominal aorta. The abdominal muscle and skin layers were successively sutured and anesthesia was reversed (atipamezole, 1 mg kg$^{-1}$; i.m.). For postoperative analgesia, the animals received carprofen (4 mg kg$^{-1}$ d$^{-1}$; i.p.) for 2 days and and were allowed to recover for at least 7 days.

**Exercise model**. Exercise capacity of experimental rats was determined using a single lane rodent treadmill (Harvard Apparatus) as described[47]. The animals were selected on the basis of their exercise compliance and subjected to daily recruitment/training sessions involving running speeds of 20–30 cm s$^{-1}$ over a 5 min period after 15 min of acclimatization to the treadmill environment. To determine the exercise capacity, treadmill speed was raised from 25 cm s$^{-1}$ in increments of 5 cm s$^{-1}$ every 5 min until the humanely defined point of exhaustion. Experiments were conducted by an investigator blinded to the nature of the experimental groups. The distance covered by the animal was recorded and exercise capacity was expressed as work done in Joules (kg m$^2$ s$^{-2}$).

**Histology and immunohistochemistry**. At the end of the experiments, the rats were given an anesthetic overdose (pentobarbitone sodium, 200 mg kg$^{-1}$, i.p.), perfused transcardially with 4% paraformaldehyde in 0.1 M phosphate buffer (pH 7.4), and post-fixed in the same solution for 4–5 days at 4 °C. After cryoprotection in 30% sucrose, serial transverse sections (30–40 µm) of the medulla oblongata were cut using a freezing microtome. After antigen retrieval in 1% citrate buffer at 80 °C, free-floating tissue sections were incubated with chicken anti-GFP (1:250; Aves Labs, Cat. GFP-1020), rabbit anti-GFAP (1:1000; DAKO, Cat. z-0334), mouse anti-NeuN (1:1000; EMD Millipore, Cat. MAB377), and/or goat anti-ChAT antibody (1:200, EMD Millipore, Cat. AB144P) overnight at 4 °C. The sections were subsequently incubated in specific secondary antibodies conjugated to the fluorescent probes (each 1:250; Life Science Technologies) for 1 h at room temperature. Images were obtained with a confocal microscope (Zeiss LSM 510).

**Data analysis**. The respiratory cycle duration ($T_{TOT}$) was measured for each respiratory cycle after the animals had habituated to the plethysmography chamber environment for at least 60 min. The average $T_{TOT}$ calculated for the periods of calm wakefulness and/or quiet sleep recorded in a 30-min period following acclimatization to the chamber environment was used to determine the resting respiratory frequency (number of breaths per minute, $f_R$). The frequency distribution of the instantaneous rate of all respiratory-related events (including signing and sniffing) in the 30-min assay period was analyzed, plotted, and reported as averages for each of the experimental groups. Poincaré plots of $T_{TOT}$ for the $n$th cycle vs. $T_{TOT}$ for the $n$th+1 cycle were used to evaluate the temporal

dispersion of $T_{TOT}$. Variability of $T_{TOT}$ was determined as described previously[48]. Tidal volume ($V_T$, normalized to the body weight) was determined by measuring the pressure changes in the chamber. Calculated values of minute ventilation ($V_E = f_R \times V_T$) were averaged and reported in arbitrary units. In addition to quantifying $f_R$, we also determined the frequency of sighs—augmented breaths that occur on top of normal inspirations[18,49]. A sigh was defined as a high-amplitude, biphasic augmented inspiratory breath (Supplementary Fig. 3) that started near the peak of a normal inspiration and lasted for a period that exceeded the duration of the previous inspiration[49,50]. Sighs were also recognizable by the lengthening of the respiratory cycle (i.e., increase in $T_{TOT}$) immediately after the sigh (Supplementary Fig. 3). Sigh frequency was calculated (and verified manually) offline using *Spike2* software as the frequency of augmented breaths with $V_T$ that was at least two times larger than the mean $V_T$ and a $T_{TOT}$ that was >50% longer than the average $T_{TOT}$ of the previous five cycles. In anesthetized rats, sigh was defined as a burst of phrenic nerve discharge with an amplitude of >50% higher that the preceding eupneic breath, followed by a brief period of apnea.

In box and whisker plots, the central black dot illustrates the mean, the central line shows the median, the edges of the box define the upper and lower quartile values, and whiskers show the minimum-maximum range of the data.

The data were compared using nonparametric Mann–Whitney U by ranks test, Wilcoxon matched-pairs signed-rank test, Kruskal–Wallis analysis of variance by ranks followed by Dunn's post hoc test, as appropriate. Differences with $p < 0.05$ were considered to be significant.

**Data availability**. The data that support the findings of this study are available from the corresponding authors upon request.

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

## Acknowledgements

This work was supported by The Wellcome Trust (A.V.G.), British Heart Foundation (A.V.G., Ref. RG/14/4/30736), BBSRC (S.K., Refs. BB/L019396/1 and BB/K009192/1), the Medical Research Council (S.K., Ref. MR/L020661), and in part by the Intramural Research Program of the NIH, NINDS. A.V.G is a Wellcome Trust Senior Research Fellow (Refs. 095064 and 200893). S.S. is an NIH-UCL GPP Fellow. We thank Professor Shamshad Cockcroft for her help with PLC activity assay and Professor Philip Haydon for providing dnSNARE construct. We are grateful to Professor David Attwell and Dr. Richard D. Fields for their comments on an earlier version of the manuscript.

## Author contributions

A.V.G. and J.C.S. designed research; S.S., E.A.T., P.S.H., A.H., S.M.T., B.L., and N.M. performed research; A.G.T. and S.K. contributed unpublished reagents/analytic tools; S.S., E.A.T., P.S.H., A.H., and S.M.T. analyzed data; A.V.G. and S.S. wrote the paper. S.K. and J.C.S. revised the article critically for important intellectual content.

## Additional information

**Competing interests:** The authors declare no competing financial interests.

