## [Peer Review File · Nature Communications]

Editorial Note: This paper was previously reviewed at another journal which does not have Transparent Peer Review. This file includes only the reviewer comments and author responses while at Nature Communications.

Reviewers' comments:

Reviewer #3 (Remarks to the Author):

In this manuscript, Gourine's group have investigated the involvement of brain stem astrocytes in the generation of the respiratory rhythm in response to demanding exercise conditions. Authors used virus expression to disrupt vesicular release from astrocytes and DREADDs to stimulate them, and tested the consequences of astrocyte manipulation in breathing in vivo. Authors' major conclusion is that astrocytes modulate the resting activity of circuits generating the respiratory rhythm, and that they contribute to the respiratory responses in conditions of increased metabolic demand.

This is an interesting study that adds valuable information regarding astrocyte-neuron interaction, a relevant and emerging, yet debated, topic in neuroscience.

The manuscript presents a collection of interesting observations using a plethora of approaches. However, in some instances, the proposed mechanistic interpretation is weakly supported by experimental evidence. Specific conclusions seem to be based on circumstantial evidence, and, in several cases, on previous related papers published by the authors. Overall, the results are consistent with the involvement of astrocytes in the respiratory regulation, but considered individually, the experimental evidence clearly needs to be strengthened to support the mechanisms proposed.

In summary, I think that present manuscript is of potential value to be published in this prestigious journal, but several concerns need to be addressed.

Specific comments

Figure 1 b shows that the number of docked vesicles is significantly reduced in astrocytes expressing dnSNARE. This is surprising because dnSNARE is supposed to impair SNARE-dependent vesicle fusion, whereas vesicle trafficking and docking are not expected to be affected. How authors explain such effect? Is this also observed in TeLC treated cells? Authors should, provide some evidence to discard uncontrolled unspecific alterations.

Conceding that the number of docked vesicles is reduced through unknown mechanisms, the number of fusion events is not surprising to be reduced, since less vesicles are docked. However, this effect would not necessarily indicate that vesicle fusion is impaired, rather, it could be simply accounted for the reduced number of available vesicles. Again, this would indicate an effect mediated by an alternative mechanism as proposed. This is worrisome because it may indicate unspecific alterations of astrocyte cell biology.

Authors propose that DREADDs are constitutively active. This is an important conclusion that needs to be properly supported. The provided observation is consistent with the hypothesis, but there is no experimental evidence directly testing the hypothesis. For any type of receptor, the adequate manner to test the hypothesis would be using antagonists of the receptors. The fact that there is no such antagonist does not allow to reach such important conclusion. Alternative interpretations can explain the observations, such as enhanced astrocytic reactivity induced by virus expression. Therefore, authors cannot conclude that there is a constitutive activity of DREADDs unless they provide solid

experimental evidence, rather than circumstantial observation, to directly test the idea.

Related to the previous point, the fusion events reported in Fig. S7f notably show similar effects in unstimulated Dredd- and CatCh-expressing cells. This suggests a non-specific activation of astrocytes transfected with virus.

Notably, the claimed constitutive activity of DREADDs, which cannot be accepted unless properly tested, seems to be unnecessary hypothesis in the present study because Dredd activation by CNO produced a large effect.

Reviewer #4 (Remarks to the Author):

Summary:

The manuscript by Sheikhabahaei et al. is the first attempt to address an important and unanswered question: does signaling from preBötC astrocytes contribute to basal respiratory rhythm generation? To answer this question, the authors use two constructs to inhibit synaptic vesicle release in preBötC astrocytes, and in doing so, observe decreased respiratory and sigh rates in consciously breathing rats in normoxia and hypoxia and decreased respiratory rate in hypercapnia and during exercise. In a complementary experiment, they increase vesicle release by expression of a modified G-protein coupled receptor (DREADD) and observe an increase in respiratory rate. This leads them to propose that astrocytes release ATP which increases the frequency of preBötC rhythm generation. The manuscript is well written and concise; however, several results need more analysis and clarification and several technical concerns remain.

1. Methodology to analyze respiration in conscious animals. The primary reported change in respiration after TeLC or dnSNARE viral injection is a change in the basal respiratory frequency. However, it is unclear from the methods how the respiratory rate is calculated and a major concern is that significant variability in the result can occur depending on the method used. For example, if the rate is calculated from breathing during sleep vs. calm wakefulness vs. actively sniffing, it will be immensely different. There is so much information in the breathing pattern from awake animals and the average breathing rate is a poor measure of this.

Unlike what is reported in Figure 1, it appears from the data in Supplemental Figure 4 that the primary change in breathing after TeLC and dnSNARE virus injection is the regularity of the rhythm. If the basal breathing rate was slower, we would expect the points in the scatter plot in Supplemental Figure 4 panel A to be shifted along the diagonal (upward and rightward). Further support against a change in basal preBötC rhythmicity comes from Supplemental Figure 6D and a recently published manuscript by Rajani et al. (Release of ATP by preBötzing complex astrocytes contributes to the hypoxic ventilatory response via a Ca²⁺-dependent P2Y1 receptor mechanism. *J. Physiology*, 2017) where the reported respiratory rate under anesthesia is unchanged in dnSNARE or TeLC experimental rats.

These points suggest that perhaps the reported decreased average respiratory rate in Figure 1 is due to increased regularity in breathing in virally injected animals. This could be because they behave differently (breathing more calmly in general) or the breaths analyzed in experimental vs. control animals are not from comparable behavioral states. The authors should be more clear about their analysis in the methods, provide an analysis of the behavioral states of control vs. injected animals, and also provide a more comprehensive analysis of breathing rate in Figure 1. For example, instead of plotting the frequency, the authors should provide a kernel density plot of the instantaneous respiratory rate for each breath over a range of frequencies from 0-12Hz. If there is a shift in basal respiratory rate, then we can expect the peak of this plot to also be shifted leftward.

2. Respiratory rate analysis of TMPAP injected rats. The superficial characterization of breathing in Figure 1 that is described above is also evident in experiments where all preBötC cells are made to express TMPAP (Figure 1H). In this manuscript, the authors report a decrease in respiratory frequency, however, in a previous manuscript, Angelova et al. (Functional oxygen sensitivity of astrocytes. *J. Neurosci.* 2015; Figure 5), the same experiment (TMPAP expression in preBötC cells) is reported to cause no change in the basal respiratory rate. Furthermore, Rajani et al. (Release of ATP by preBötzing complex astrocytes contributes to the hypoxic ventilatory response via a Ca²⁺-dependent P2Y1 receptor mechanism. *J. Physiology*, 2017), reports that injection of MRS2279 (a P2Y1 receptor antagonist) does not change the basal respiratory rate. Please clarify the differences in experimental observations and explain what accounts for the new result of a decrease in respiratory rate after TMPAP injection.

3. Modulation of sighing. The authors claim that “the actions of bombesin-like peptides on preBötC circuits are potentially mediated by astrocytes”. Although they convincingly show that the sigh rate decreases after TeLC, dnSNARE and TMPAP preBötC injection, there are several discrepancies with Li et al. (The peptidergic control circuit for sighing. *Nature* 2016) that need to be clarified: 1) In Li et al., in situ hybridizations for NMBR and GRPR in the preBötC did not show widespread expression of these transcripts throughout the preBötC, which is what would be expected if all preBötC astrocytes are bombesin responsive (Supplemental Figure 6), 2) the in vivo microinjection experiment does not appear to be robust. 250uM bombesin injected into the preBötC causes a mild increase in sigh frequency (compare to <10uM for NMB or GRP in Li et al.). This is three orders of magnitude more bombesin than is required to increase calcium in astrocytes in in vitro experiments performed in Supplemental Figure 6A. The control injections have significant experimental variability (Supplemental Figure 6D) and may confound the proclaimed >60% decreased response in experimental animals.

4. DREADD induced changes in respiration. To substantiate the idea that expression of DREADD decreases the respiratory regularity by increasing release of ATP from astrocytes, the authors should demonstrate that CNO induced changes in breathing are eliminated by pharmacologically antagonizing ATP signaling in the preBötC or after coinjection of TMPAP virus.

5. Conclusion that hypoxia, hypercapnia, and exercise capacity requires astrocytic vesicle release. In Figure 1 the authors claim that the basal respiratory rate of the animals is decreased by blocking vesicle release from astrocytes. In Figure 2, they show that astrocyte vesicle release is also required for a full hypoxic and hypercapnic response. However, the conclusion that they impact changes in respiration disproportionately to the changes seen to normoxic respiration needs to be further clarified. For example, if we are to normoxic respiratory rate decreases by 11% after dnSNARE or TeLC virus injection, is the hypoxic and hypercapnic rate significantly less than 89% of the normal ventilatory response?

Additional comments:

Please label panels in Supplemental Figure 6 and correct the Y-axis in Supplemental Figure 9.

Manuscript ID: NCOMMS-17-15074-T
Responses to the referees' comments

We would like to thank all the reviewers and the Editors of Nature Neuroscience and **Nature Communications** for their time taken to evaluate our submission and overall positive assessment of our work. We are grateful for the constructive comments provided and have taken a full account of the raised criticisms. We are delighted to have another opportunity to re-submit our work. We now include additional experimental data/analysis requested by the reviewers, and provide a full response to their comments as well as a thoroughly revised manuscript.

Below we state the criticisms ("critique") and then provide our detailed responses.

Reviewer #3:

In this manuscript, Gourine's group have investigated the involvement of brain stem astrocytes in the generation of the respiratory rhythm in response to demanding exercise conditions. Authors used virus expression to disrupt vesicular release from astrocytes and DREADDs to stimulate them, and tested the consequences of astrocyte manipulation in breathing in vivo. Authors' major conclusion is that astrocytes modulate the resting activity of circuits generating the respiratory rhythm, and that they contribute to the respiratory responses in conditions of increased metabolic demand.

This is an interesting study that adds valuable information regarding astrocyte-neuron interaction, a relevant and emerging, yet debated, topic in neuroscience.

The manuscript presents a collection of interesting observations using a plethora of approaches. However, in some instances, the proposed mechanistic interpretation is weakly supported by experimental evidence. Specific conclusions seem to be based on circumstantial evidence, and, in several cases, on previous related papers published by the authors. Overall, the results are consistent with the involvement of astrocytes in the respiratory regulation, but considered individually, the experimental evidence clearly needs to be strengthened to support the mechanisms proposed.

In summary, I think that present manuscript is of potential value to be published in this prestigious journal, but several concerns need to be addressed.

Response: We would like to thank this referee for his/her time taken to review our manuscript and very positive assessment of our work. We now include additional experimental data in our revised manuscript and provide detailed responses to all the comments raised.

Critique: Figure 1 b shows that the number of docked vesicles is significantly reduced in astrocytes expressing dnSNARE. This is surprising because dnSNARE is supposed to impair SNARE-dependent vesicle fusion, whereas vesicle trafficking and docking are not expected to be affected. How authors explain such effect? Is this also observed in TeLC treated cells? Authors should, provide some evidence to discard uncontrolled unspecific alterations.

Response: We respectfully disagree with the reviewer here. The role of SNARE proteins in vesicle docking remains controversial. Although, this role has been questioned based on the EM evidence, there is significant experimental data that SNARE proteins are required for vesicle docking. See for example: <https://www.ncbi.nlm.nih.gov/pubmed/22869597> . Please also see most recent high profile paper which used this approach to block astroglial vesicular signalling pathways: <https://www.ncbi.nlm.nih.gov/pubmed/28479102> .

A comprehensive study investigated the mechanisms underlying the effect of dnSNARE expression on exocytosis in astrocytes (<https://www.ncbi.nlm.nih.gov/pubmed/27056575>).

This paper is not reporting the numbers of “docked” vesicles in astrocytes treated with dnSNARE, but shows whole cell vesicles visualized using STED/SIM. The authors state that “...the expression of dnSNARE peptide strongly reduced the occurrence of irreversible exocytotic events only... The frequency of reversible events was unchanged from that in controls”. This means that there are overall fewer events and less productive exocytosis. The authors interpret these observations as “...dnSNARE treatment locks vesicles in the transient-fusion stage, preventing the pore from widening to the full-fusion stage”. However, the authors show no evidence that the vesicles are “locked” and stay at the membrane docked. An alternative explanation is that vesicles which are not able to fuse because of dnSNARE promptly leave the membrane (“kiss and run”) and that is why we observe smaller numbers of vesicles at and near the membrane. To address this point of the reviewer we now replaced the expression “docked vesicles” with “juxtamembrane vesicles”

Critique: Conceding that the number of docked vesicles is reduced through unknown mechanisms, the number of fusion events is not surprising to be reduced, since less vesicles are docked. However, this effect would not necessarily indicate that vesicle fusion is impaired, rather, it could be simply accounted for the reduced number of available vesicles. Again, this would indicate an effect mediated by an alternative mechanism as proposed. This is worrisome because it may indicate unspecific alterations of astrocyte cell biology.

Response: Please see our reasoning above. The mechanism of dnSNARE-mediated blockade of vesicular release mechanisms in astrocytes has been described in detail previously (<https://www.ncbi.nlm.nih.gov/pubmed/27056575>). In this work we further validated this approach for the purpose of our study and report that our viral vectors successfully target brainstem astrocytes to express dnSNARE and that dnSNARE expression effectively impairs astroglial vesicular release mechanisms, as expected. Most importantly, expression of either dnSNARE or tetanus toxin light chain (TeLC) in astrocytes of the pre-Bötzing complex had similar effects on resting respiratory activity, frequency of sighs, regularity of breathing, respiratory responses to hypoxia and hypercapnia as well as exercise capacity, indicating that both approaches are specific and block the same signalling pathway, i.e. vesicular release mechanism.

Critique: Authors propose that DREADDs are constitutively active. This is an important conclusion that needs to be properly supported. The provided observation is consistent with the hypothesis, but there is no experimental evidence directly testing the hypothesis. For any type of receptor, the adequate manner to test the hypothesis would be using antagonists of the receptors. The fact that there is no such antagonist does not allow to reach such important conclusion. Alternative interpretations can explain the observations, such as enhanced astrocytic reactivity induced by virus expression. Therefore, authors cannot conclude that there is a constitutive activity of DREADDs unless they provide solid experimental evidence, rather than circumstantial observation, to directly test the idea.

Response: We respectfully disagree with the reviewer here and are unsure why he/she dismisses the experimental evidence we present which strongly suggest that DREADDq expressed in astrocytes is constitutively active (Fig. 2). In the original paper by Armbruster and colleagues (2007) the authors say “Because many of the mutants with the highest CNO potencies had high levels of constitutive activity (Fig. 2 A and SI Table 2), we next screened a focused library of hM3 receptor mutants in HEK T cells to generate a receptor that was potently activated by CNO with minimal constitutive activity” (<https://www.ncbi.nlm.nih.gov/pubmed/17360345>). Clearly all the designed receptors express a certain level of constitutive activity and we are not sure why this referee is surprised to see the evidence of this in our experiments.

In our paper we report that astrocytes expressing DREADD_{Gq} show:

- (i) higher level of PLC activity (Fig 2d);
- (ii) higher rate of spontaneous fusion of ATP containing vesicles (Fig 2e);
- (iii) facilitated release of ATP by astrocytes transduced *in vivo* (Fig 2i, Fig 2j).

In our opinion this validation is comprehensive. Increased level of PLC activity in DREADD_{Gq}-expressing cells alone is a direct evidence of constitutive G_q-mediated signalling. Numerous GPCRs (not only DREADD) are constitutively active when expressed at high levels, which is the case with the recombinant receptors expressed using viruses with strong promoters. This is discussed in numerous reviews and textbooks (e.g. "Signal Transduction" by Gampert BD et al). The main reason for this phenomenon is that GPCRs stochastically fluctuate between different conformations and the ligands stabilize their active conformation whereby they activate the relevant G proteins. Therefore, at low level of expression the frequency of spontaneous activation of G proteins is low but at high level achieved by virally-mediated expression it increases and becomes (functionally) significant.

Critique: Related to the previous point, the fusion events reported in Fig. S7f notably show similar effects in unstimulated DREADD- and CatCh-expressing cells. This suggest a non-specific activation of astrocytes transfected with virus.

Response: We thank the reviewer for this observation. There were no differences in the respiratory rate (our primary endpoint) between the naïve (not transduced) animals and groups of animals transduced to express various control transgenes by the preBötC astrocytes (Supplementary Figure 3f). Alternatively, the data questioned by the reviewer may suggest that control (or naïve) astrocytes are somehow affected by CNO. A recent high-profile paper reported that CNO in fact shows low affinity for DREADDs and the effects of CNO are largely attributed to its rapid conversion to clozapine which shows high DREADD affinity and potency (<https://www.ncbi.nlm.nih.gov/pubmed/28774929>). Clozapine interacts with 5-HT_{2A} receptors which astrocytes do express (e.g. <https://www.ncbi.nlm.nih.gov/pubmed/9542727>). Therefore, although astrocytes expressing DREADD_{Gq} can be activated by application of CNO, this approach has its drawbacks. In our paper we report data suggesting that commonly used mutant of DREADD_{Gq} is constitutively active when expressed in astrocytes and this constitutive activity can be harnessed to study the functional role of astrocytes as complementary, gain-of-function experimental approach.

Critique: Notably, the claimed constitutive activity of DREADDs, which cannot be accepted unless properly tested, seems to be unnecessary hypothesis in the present study because DREADD activation by CNO produced a large effect.

Response: Please see our responses above. In the revised version of the manuscript we now report new experimental data (Fig 2k) showing that the stimulatory effect of DREADD_{Gq} expression in preBötC astrocytes on the respiratory activity is effectively blocked by co-expression of an ATP-degrading enzyme TMPAP. These data suggest that the effect of constitutive DREADD_{Gq} activity on breathing is specific and mediated by the release and actions of purines on the respiratory network.

Reviewer #4

Summary: The manuscript by Sheikhabahaei et al. is the first attempt to address an important and unanswered question: does signaling from preBötC astrocytes contribute to basal respiratory rhythm generation? To answer this question, the authors use two constructs to inhibit synaptic vesicle release in preBötC astrocytes, and in doing so, observe decreased respiratory and sigh rates in consciously breathing rats in normoxia and hypoxia and decreased respiratory rate in hypercapnia and during exercise. In a complementary experiment, they increase vesicle release by expression of a modified G-protein coupled receptor (DREADD) and observe an increase in respiratory rate. This leads them to propose that astrocytes release ATP which increases the frequency of preBötC rhythm generation. The manuscript is well written and concise; however, several results need more analysis and clarification and several technical concerns remain.

Response: We would like to thank this referee for his/her time taken to review our manuscript and overall positive assessment of our work. We now include additional experimental data in our revised manuscript and provide detailed responses to all the comments raised.

Critique: 1. Methodology to analyze respiration in conscious animals. The primary reported change in respiration after TeLC or dnSNARE viral injection is a change in the basal respiratory frequency. However, it is unclear from the methods how the respiratory rate is calculated and a major concern is that significant variability in the result can occur depending on the method used. For example, if the rate is calculated from breathing during sleep vs. calm wakefulness vs. actively sniffing, it will be immensely different. There is so much information in the breathing pattern from awake animals and the average breathing rate is a poor measure of this.

Response: We thank the reviewer for this comment. In our experiments, the animals were placed in the plethysmography chamber and were allowed to acclimatize to the chamber environment for ~60 min. The respiratory activity was recorded for 30 min and the resting breathing rate was calculated for the periods of calm wakefulness and/or quiet sleep. We now indicate this in the revised version of the manuscript. In accord with the referee's request we now re-analyzed all the data and in the revised manuscript now report frequency distribution of all respiratory-related events (including signing and sniffing) in the 30-min assay period in animals expressing CatCh control, dnSNARE (Fig 1g), TeLC (Fig 1h), TMPAP (Fig 1i) or DREADDGq (Fig 2k) in preBötC astrocytes.

Critique: Unlike what is reported in Figure 1, it appears from the data in Supplemental Figure 4 that the primary change in breathing after TeLC and dnSNARE virus injection is the regularity of the rhythm. If the basal breathing rate was slower, we would expect the points in the scatter plot in Supplemental Figure 4 panel A to be shifted along the diagonal (upward and rightward). Further support against a change in basal preBötC rhythmicity comes from Supplemental Figure 6D and a recently published manuscript by Rajani et al. (Release of ATP by preBötC astrocytes contributes to the hypoxic ventilatory response via a Ca²⁺-dependent P2Y₁ receptor mechanism. *J. Physiology*, 2017) where the reported respiratory rate under anesthesia is unchanged in dnSNARE or TeLC experimental rats.

Response: We believe that these differences are due to the use of anesthetics in that study. The data reported in Supplemental Figure 6D (revised Supplemental Fig. 5c) and the data reported by Rajani and colleagues (<https://www.ncbi.nlm.nih.gov/pubmed/28678385>) were obtained in anaesthetized animals and not directly comparable with the results obtained in unanaesthetized animals (reported in this manuscript). It is not surprising that moderate differences (10-15%) in resting respiratory rate are no longer observed when the animals are anaesthetized.

Critique: These points suggest that perhaps the reported decreased average respiratory rate in Figure 1 is due to increased regularity in breathing in virally injected animals. This could be because they behave differently (breathing more calmly in general) or the breaths analyzed in

experimental vs. control animals are not from comparable behavioral states. The authors should be more clear about their analysis in the methods, provide an analysis of the behavioral states of control vs. injected animals, and also provide a more comprehensive analysis of breathing rate in Figure 1. For example, instead of plotting the frequency, the authors should provide a kernel density plot of the instantaneous respiratory rate for each breath over a range of frequencies from 0-12Hz. If there is a shift in basal respiratory rate, then we can expect the peak of this plot to also be shifted leftward.

Response: We thank the reviewer for this excellent comment. We now re-analyzed all the data and in the revised manuscript now report frequency distribution of all respiratory-related events (including signing and sniffing) in 30-min assay in animals expressing CatCh control, dnSNARE (Fig 1g), TeLC (Fig 1h), TMPAP (Fig 1i) or DREADD_{Gq} (Fig 2k) in preBötC astrocytes. In accord with the referee's prediction, expression of either dnSNARE, TeLC or TMPAP was associated with a clear leftwards shift in peak respiratory frequency distribution (Figs. 1g-i). DREADD_{Gq} expression in preBötC astrocytes resulted in a higher prevalence of high-frequency events (Fig 2k).

Critique: 2. Respiratory rate analysis of TMPAP injected rats. The superficial characterization of breathing in Figure 1 that is described above is also evident in experiments where all preBötC cells are made to express TMPAP (Figure 1H). In this manuscript, the authors report a decrease in respiratory frequency, however, in a previous manuscript, Angelova et al. (Functional oxygen sensitivity of astrocytes. *J. Neurosci.* 2015; Figure 5), the same experiment (TMPAP expression in preBötC cells) is reported to cause no change in the basal respiratory rate. Furthermore, Rajani et al. (Release of ATP by preBötC astrocytes contributes to the hypoxic ventilatory response via a Ca²⁺-dependent P2Y1 receptor mechanism. *J. Physiology*, 2017), reports that injection of MRS2279 (a P2Y1 receptor antagonist) does not change the basal respiratory rate. Please clarify the differences in experimental observations and explain what accounts for the new result of a decrease in respiratory rate after TMPAP injection.

Response: We thank the reviewer for this comment and in the revised manuscript describe the analysis of the respiratory data in more detail. In our previous study mentioned by the reviewer (Angelova et al., 2015; <https://www.ncbi.nlm.nih.gov/pubmed/26203141>) the experiments aimed to determine the role of astrocytes in central respiratory sensitivity to hypoxia and studies involving TMPAP expression in the brainstem were performed in rats with denervated carotid bodies (10 weeks after peripheral chemodenervation). The data reported by Rajani and colleagues (<https://www.ncbi.nlm.nih.gov/pubmed/28678385>) were obtained in anaesthetized animals and the experimental design involved unilateral injections of a P2Y1 receptor antagonist. Here we targeted our injections to the preBötC bilaterally and recorded the respiratory activity in un-anaesthetized animals with intact peripheral chemoreceptors, therefore, these new data are not directly comparable with the results reported previously.

Critique: 3. Modulation of sighing. The authors claim that "the actions of bombesin-like peptides on preBötC circuits are potentially mediated by astrocytes". Although they convincingly show that the sigh rate decreases after TeLC, dnSNARE and TMPAP preBötC injection, there are several discrepancies with Li et al. (The peptidergic control circuit for sighing. *Nature* 2016) that need to be clarified: 1) In Li et al., in situ hybridizations for NMBR and GRPR in the preBötC did not show widespread expression of these transcripts throughout the preBötC, which is what would be expected if all preBötC astrocytes are bombesin responsive (Supplemental Figure 6), 2) the in vivo microinjection experiment does not appear to be robust. 250uM bombesin injected into the preBötC causes a mild increase in sigh frequency (compare to <10uM for NMB or GRP in Li et al.). This is three orders of magnitude more bombesin than is required to increase calcium in astrocytes in in vitro experiments performed in Supplemental Figure 6A. The control injections have significant experimental variability (Supplemental Figure 6D) and may confound the proclaimed >60% decreased response in experimental animals.

Response: We thank the reviewer for this comment. Please note that determining the preBötC cellular targets of bombesin-related peptides was not the main goal of this study. These

experiments were motivated by our observation of reduced sigh rate in conditions when signalling pathways in preBötC astrocytes are compromised by virally-driven expression of TeLC, dnSNARE or TMPAP, and earlier reports from the Ramirez's group on the role of purinergic signalling in sigh generation. We agree that without significant additional experimental work these results are not sufficient to draw a firm conclusion on bombesin cellular targets within the preBötC, therefore, the data are only shown in the Online Supplement (revised Supplemental Fig. 5). Yet, we believe that these results are important to guide/facilitate future research as they demonstrate that brainstem astrocytes (at least in culture) express receptors which can be activated by bombesin-related peptides and the effect of bombesin on sigh frequency is reduced when astroglial signalling mechanisms are blocked by dnSNARE expression. Even the original comprehensive study by Li and colleagues (PMC: 4852886) did not report which cells in the preBötC are actually expressing NMBR and GRPR. It is possible that the astrocyte expression is low and limited to astroglial process which is closely adjacent to NMB and/or GRP containing projections (there is general understating in the field that astrocytes are "tuned" to monitor local neurochemical environment). It is also possible that NMBR and GRPR are only expressed by preBötC neurons, but both inputs ("bombesin-ergic" and purinergic) are required for sigh generation. Answering these intriguing questions, however, is beyond the scope of our study.

For the *in vivo* experiments we used the protocol described in the first publication by the Feldman's group (<https://www.ncbi.nlm.nih.gov/pubmed/23719793>). The authors used bombesin in a concentration of 240 μM injected into the preBötC bilaterally at a volume of 90 nl/side. We injected 250 μM bombesin solution unilaterally at a volume of 50 nl and recorded significant increases in the sigh rate. Why higher concentration of bombesin is required is unclear but may reflect higher affinity of the mammalian receptors to NMB/GRP or simply higher chemical purity of the commercially available NMB and GRP peptides. Also, it is usually difficult to compare the concentrations of a particular substance used for the *in vivo* and *in vitro* experiments. Very low volumes of concentrated solutions are usually applied *in vivo* to achieve local tissue concentration of a substance comparable to that applied in a bath in the *in vitro* studies.

We agree that control injections produced variable responses which may reflect slight differences in the injections sites placed within the preBötC. However, the effect of bombesin-related peptides on sigh generation appears to be inherently variable as described in the original paper by Li and colleagues (please examine Figure 2d in *Nature* 530: 293, 2016). Nonetheless, in our experiments these bombesin-induced responses were markedly reduced when preBötC astrocytes were transduced to express dnSNARE. To address this point of the reviewer we now modified the text of the paper to read:

"Blockade of vesicular release mechanisms in preBötC astrocytes (dnSNARE expression) significantly reduced the effect of bombesin on sigh frequency (Supplementary Fig. 5c), suggesting that the actions of bombesin-like peptides on preBötC circuits¹⁴ are potentially mediated by astrocytes".

For this resubmission we also performed additional experiments and now report new data showing that the effect of bombesin on $[\text{Ca}^{2+}]_i$ in astrocytes is abolished in the presence of a highly selective neuromedin B receptor antagonist BIM 23042 (Supplementary Fig. 5a), confirming specificity of bombesin effect on astrocytes.

Critique: 4. DREADD induced changes in respiration. To substantiate the idea that expression of DREADD decreases the respiratory regularity by increasing release of ATP from astrocytes, the authors should demonstrate that CNO induced changes in breathing are eliminated by pharmacologically antagonizing ATP signaling in the preBötC or after coinjection of TMPAP virus.

Response: We thank the reviewer for this comment. For this resubmission we have evaluated the effect of DREADDGq and TMPAP co-expression in the preBötC on the respiratory activity. These new data are presented in Fig. 2k. The enzymatic activity of TMPAP completely abolished

the stimulatory effect of DREADDGq expression in preBötC astrocytes on resting respiratory rate. Co-expression of DREADDGq and TMPAP was associated with a significant reduction of the respiratory rate below the baseline – the effect similar to that induced by TMPAP expression alone. These data strongly support our earlier observations that TMPAP is indeed highly effective in blocking ATP-mediated signalling.

Critique: 5. Conclusion that hypoxia, hypercapnia, and exercise capacity requires astrocytic vesicle release. In Figure 1 the authors claim that the basal respiratory rate of the animals is decreased by blocking vesicle release from astrocytes. In Figure 2, they show that astrocyte vesicle release is also required for a full hypoxic and hypercapnic response. However, the conclusion that they impact changes in respiration disproportionately to the changes seen to normoxic respiration needs to be further clarified. For example, if we are to normoxic respiratory rate decreases by 11% after dnSNARE or TeLC virus injection, is the hypoxic and hypercapnic rate significantly less than 89% of the normal ventilatory response?

Response: We thank the reviewer for this comment. The text of the revised manuscript has been modified to read:

*“In conscious rats, bilateral expression of dnSNARE or TeLC in preBötC astrocytes (Fig. 1f; Supplementary Fig. 2,3) resulted in a significant reduction in resting breathing frequency (f_R) by **11%** ($94 \pm 2 \text{ min}^{-1}$ vs $106 \pm 5 \text{ min}^{-1}$ in controls; $n=5$, $p=0.016$) and by **11%** ($92 \pm 2 \text{ min}^{-1}$ vs $103 \pm 3 \text{ min}^{-1}$ in controls; $n=12$, $p=0.011$), respectively (Figs. 1g,h)”.*

*“Expression of dnSNARE attenuated hypoxia-induced increases in f_R by **27%** ($159 \pm 10 \text{ min}^{-1}$ vs $217 \pm 7 \text{ min}^{-1}$ in controls; Fig. 4a) and in minute ventilation by **34%** (Fig. 4a). TeLC expression had a similar effect (Supplementary Fig. 6), consistent with a proposed role of astrocytes as CNS oxygen sensors¹⁰”.*

*“In conscious rats, bilateral expression of dnSNARE or TeLC in preBötC astrocytes reduced the f_R responses to hypercapnia (6% inspired CO_2) by **23%** ($141 \pm 6 \text{ min}^{-1}$ vs $182 \pm 3 \text{ min}^{-1}$ in controls; $n=5$, $p=0.008$) and **20%** ($151 \pm 6 \text{ min}^{-1}$ vs $190 \pm 8 \text{ min}^{-1}$ in controls; $n=9$, $p=0.005$), respectively (Fig. 4b), concomitantly reducing minute ventilation (Fig. 4b)”.*

Also, please review revised Fig. 4a and Supplementary Fig. 6. Although, expression of dnSNARE or TeLC in preBötC astrocytes reduced resting respiratory rate, minute ventilation at normoxia was similar to that in animals expressing control transgene. Marked differences in ventilation between the experimental and control groups were only observed during the hypoxic challenge.

Critique: Additional comments:

Please label panels in Supplemental Figure 6 and correct the Y-axis in Supplemental Figure 9.

Response: Thank you, done.

REVIEWERS' COMMENTS:

Reviewer #1 (Remarks to the Author):

Regarding the concerns expressed about the previous version, authors have provided a convincing list of reasonable arguments to support their claims, and I am satisfied with the reply.

I would only have one suggestion that the authors may want to consider: the arguments provided could be easily included as part of the discussion. They would not enlarge too much the manuscript and I believe they will be helpful for the reader.

I have no other comments and further concerns.

I reaffirm my previous assessment of the work:

This is an interesting study that adds valuable information regarding astrocyte-neuron interaction, a relevant and emerging, yet debated, topic in neuroscience.

Reviewer #2 (Remarks to the Author):

Thank you for providing the new experimental data and re-analysis of previously acquired data. They have certainly made the manuscript stronger.

The only remaining concern is the manuscript does not discuss the alternative conclusion: that the various perturbations to astrocytes could be generally changing breathing, for example by decreasing neural health and excitability. Although the authors do not favor this interpretation, none of the experiments exclude it and I fear that only including one interpretation will mislead the audience. While the authors models can certainly still be included in the discussion, please also include a discussion of the alternative and simpler interpretation of the data.

The authors claim or elude that astrocytes regulate respiratory rate and regularity, mediate sighing, and are important sensors in the hypoxic and hypercapnic respiratory response. These roles are all distinct upstream mechanisms: by sensing bombesin, hypoxia, and hypercapnia and they all funnel through the same downstream mechanism, ATP release. However, the data presented are insufficient to make this conclusion and it certainly seems confusing that general ATP release could cause all of these specific changes in breathing.

The same data presented is also consistent with the conclusion that vesicle release from astrocytes is required for normal preBötC rhythm and pattern generation, and in the absence of this, general respiratory behavior is changed (thus decrease sighing and breathing frequency under different respiratory stimuli).

Specific example that are troubling include:

- 1) The conclusion that ATP release from astrocytes maintains normal respiratory rate. However, experiments blocking P2Y1 receptors in the preBötC do not change respiratory rate (Rajani et al. 2017). Although this is under anesthesia, it suggests that ATP is not required for normal respiratory rate, unless this signaling is absent under anesthesia. A possible re-interpretation of the overexpression of TMPAP could be that ATP maintains normal astrocytic waves which are required for general astrocyte vesicle release (Scemes et al. (2006) Astrocyte calcium waves. *Glia*). Without showing that astrocytes are functioning normally, the conclusion that blocking ATP signaling in the preBötC is specific to neurons is unsupported.

2) Astrocytes mediate sighing. How is it possible that they mediate sighing when the sigh rate goes from 30/hr to 20/hr? Blocking astrocyte vesicle release does not completely eliminate sighing (unlike blocking NMBR and GRPR signaling). Additionally, the bombesin response of cultured astrocytes is blocked by NMBR antagonists, however, GRP can robustly induce sighing in the preBötC and is required by the preBötC for normal sighing. There are certainly just too many holes to attempt to make or elude to this conclusion.

3) Hypoxia and hypercapnia response. Although the reported response to hypoxia and hypercapnia is certainly blunted, there is still a robust change in tidal volume and frequency. I understand that the authors are not arguing that astrocytes completely mediate these responses, but the data could also be re-interpreted as astrocytes do not have a specific role in sensing hypercapnia and hypoxia and instead the same general change in breathing frequency during normoxia is ALSO occurring in different blood gas states, such as hypoxia and hypercapnia.

Manuscript ID: NCOMMS-17-15074A
Responses to the referees' comments

We are extremely grateful for the constructive comments of both reviewers and the Editor of **Nature Communications** and have taken full account of the raised criticisms. We are absolutely delighted that our work has been judged potentially suitable for publication. We now provide a full response to the remaining comments of both reviewers and submit the third revision of our manuscript.

Below we state the criticisms ("critique") and then provide our detailed responses.

Reviewer #1:

Regarding the concerns expressed about the previous version, authors have provided a convincing list of reasonable arguments to support their claims, and I am satisfied with the reply. I would only have one suggestion that the authors may want to consider: the arguments provided could be easily included as part of the discussion. They would not enlarge too much the manuscript and I believe they will be helpful for the reader. I have no other comments and further concerns. I reaffirm my previous assessment of the work: This is an interesting study that adds valuable information regarding astrocyte-neuron interaction, a relevant and emerging, yet debated, topic in neuroscience.

Response: We would like to thank this referee for his/her time taken to review our manuscript and very positive assessment of our work. In the revised text of the paper we now briefly discuss all the issues which were raised by this referee in previous rounds of review. These include: (i) mechanisms of dnSNARE-mediated blockade of vesicular release mechanisms in astrocytes; (ii) issue of apparent constitutive activity of DREADD_{Gq} expressed in astrocytes; and (iii) potential drawbacks in using CNO to study astrocytes.

Reviewer #2:

Thank you for providing the new experimental data and re-analysis of previously acquired data. They have certainly made the manuscript stronger.

Response: We would like to thank this referee for his/her time taken to review our manuscript and overall positive assessment of our work. Below we provide detailed responses to all the remaining concerns.

The only remaining concern is the manuscript does not discuss the alternative conclusion: that the various perturbations to astrocytes could be generally changing breathing, for example by decreasing neural health and excitability. Although the authors do not favor this interpretation, none of the experiments exclude it and I fear that only including one interpretation will mislead the audience. While the authors models can certainly still be included in the discussion, please also include a discussion of the alternative and simpler interpretation of the data.

Response: We respectfully disagree with the reviewer here. Genetic approaches we used in this study to block vesicular release by astrocytes or activate astroglial Ca²⁺ signalling pathways are very specific. If TeLC or dnSNARE expression in astrocytes would have a significant impact on health and excitability of neighbouring neurons, then we would expect to observe a much more severe breathing deficit when these transgenes are expressed within the respiratory rhythm generating circuits of the preBötC. In rats, loss of only ~600 preBötC neurons bilaterally is associated with long apneas and severe ataxic breathing pattern (PMID: 11528424). In the reduced preparations (rhythmic brainstem slice of neonatal mice), ablation of only 15% of rhythmogenic preBötC neurons is sufficient to abolish the inspiratory rhythm (PMID: 25027440). We targeted astrocytes of the whole bilateral preBötC region to express TeLC or dnSNARE and observed a relatively moderate effect on the respiratory frequency (decrease by ~10%) and no effect on minute ventilation at rest suggesting that the health of neurons which

constitute the respiratory circuits is unlikely to be affected. The physiological role of the targeted pathway/mechanism became apparent when the system was challenged, in this case when the respiratory responses of conscious animals were assessed in conditions of increased metabolic demand (hypoxia, hypercapnia or exercise).

Critique: The authors claim or elude that astrocytes regulate respiratory rate and regularity, mediate sighing, and are important sensors in the hypoxic and hypercapnic respiratory response. These roles are all distinct upstream mechanisms: by sensing bombesin, hypoxia, and hypercapnia and they all funnel through the same downstream mechanism, ATP release. However, the data presented are insufficient to make this conclusion and it certainly seems confusing that general ATP release could cause all of these specific changes in breathing. The same data presented is also consistent with the conclusion that vesicle release from astrocytes is required for normal preBötC rhythm and pattern generation, and in the absence of this, general respiratory behavior is changed (thus decrease sighing and breathing frequency under different respiratory stimuli).

Response: We respectfully disagree with the reviewer here and based on the data presented would argue that astrocytes are capable of sensing/integrating distinct metabolic and humoral stimuli and, via the release of ATP, modulate the activity of the intermingled neuronal circuits of the preBötC. PreBötC circuits generate the basic rhythm of breathing which is modulated by a variety of inputs (including signalling molecules released by neighbouring astrocytes) and then transmitted into an appropriate pattern of respiratory activity.

We are not sure why this reviewer is surprised that astrocytes and “general ATP release could cause all of these specific changes in breathing”. Let’s consider, for example, the functional role played by the carotid body. Type I (glomus) cells of the carotid body are sensitive to hypoxia, hypercapnia and various humoral factors (e.g. glucose, inflammatory mediators, etc). Upon activation, type I cells release ATP(!) to activate afferent fibers of the carotid sinus nerve to trigger adaptive changes in breathing. Removal of the carotid body reduces the respiratory frequency at rest, increases the variability of breathing and has a major impact on the ventilatory responses to hypoxia and hypercapnia. We would argue that this critical function of the peripheral respiratory chemoreceptors is to a certain extent duplicated in the CNS, with preBötC astrocytes playing an analogous role.

Critique: 1) The conclusion that ATP release from astrocytes maintains normal respiratory rate. However, experiments blocking P2Y1 receptors in the preBötC do not change respiratory rate (Rajani et al. 2017). Although this is under anesthesia, it suggests that ATP is not required for normal respiratory rate, unless this signaling is absent under anesthesia.

Response: The data reported by Rajani and colleagues (PMID: 28678385) were obtained in anaesthetized animals and the experimental design involved unilateral injections of a relatively specific P2Y₁ receptor antagonist MRS 2279. Here we targeted our injections to the preBötC bilaterally and recorded the respiratory activity in un-anaesthetized animals with intact peripheral chemoreceptors, therefore, these new data are not directly comparable with the results reported previously. Moreover, in the present study we targeted not a specific receptor subtype, but interfered with upstream mechanisms, by either blocking vesicular release in astrocytes or promoting rapid degradation of the released ATP. Our earlier studies demonstrated that in addition to P2Y₁ receptors, respiratory neurons within the rhythm generating circuits of the preBötC express other ATP receptors (e.g. P2X₂; PMID: 12878756), which are not sensitive to blockade by MRS 2279 used in the study by Rajani and colleagues.

Critique: A possible re-interpretation of the overexpression of TMPAP could be that ATP maintains normal astrocytic waves which are required for general astrocyte vesicle release (Scemes et al. (2006) Astrocyte calcium waves. *Glia*). Without showing that astrocytes are functioning normally, the conclusion that blocking ATP signaling in the preBötC is specific to neurons is unsupported.

Response: We do not quite understand this comment of the reviewer. In order to study the physiological/functional significance of a particular mechanism or process, experimental tools are designed to specifically block/inhibit this particular mechanism/process. If astrocyte function, such as generation of calcium waves that may be required for vesicular release (PMID: 17504911) is compromised by disrupting ATP-mediated signaling, and there are concomitant perturbations of breathing frequency (which must occur by affecting neuronal excitability), then this is also consistent with our general conclusions that ATP-mediated signaling is playing an important role. If we follow the reviewers' line of reasoning, we would have to propose that other astrocyte-released signaling molecules may also be involved in affecting neuronal excitability, which we do not rule out.

Critique: Astrocytes mediate sighing. How is it possible that they mediate sighing when the sigh rate goes from 30/hr to 20/hr? Blocking astrocyte vesicle release does not completely eliminate sighing (unlike blocking NMBR and GRPR signaling). Additionally, the bombesin response of cultured astrocytes is blocked by NMBR antagonists, however, GRP can robustly induce sighing in the preBötC and is required by the preBötC for normal sighing. There are certainly just too many holes to attempt to make or elude to this conclusion.

Response: We agree and in our paper we are not claiming that 'astrocytes mediate sighing'. We say in the text that "*sigh generation may be modulated by signaling molecules released by preBötC astrocytes in response to various stimuli, including locally released bombesin-like peptides*". As we argued in our previous response letter, determining the preBötC cellular targets of bombesin-related peptides was not the main goal of this study. We observed and reported the effect of bombesin on Ca^{2+} in brainstem astrocytes and the effect of compromised preBötC astroglial vesicular release mechanisms on bombesin-induced increases in sigh frequency. Even the original comprehensive study by Li and colleagues (PMC: 4852886) did not report which cells in the preBötC are actually expressing NMBR and GRPR. A recent study reported that the majority of retrotrapezoid nucleus neurons express NMB (PMID: 29066557). These cells project to the preBötC and are strongly activated by CO_2 . Yet, the effects of systemic hypercapnia on sigh rate are usually very modest. Hence, we agree that there are "too many holes" in this story (not just in our small contribution to it). Therefore, we report our observations in the Supplementary Online Material and hope they may facilitate future research in this field.

Critique: Hypoxia and hypercapnia response. Although the reported response to hypoxia and hypercapnia is certainly blunted, there is still a robust change in tidal volume and frequency. I understand that the authors are not arguing that astrocytes completely mediate these responses, but the data could also be re-interpreted as astrocytes do not have a specific role in sensing hypercapnia and hypoxia and instead the same general change in breathing frequency during normoxia is ALSO occurring in different blood gas states, such as hypoxia and hypercapnia.

Response: Please review Fig. 4a and Supplementary Fig. 6. Although, expression of dnSNARE or TeLC in preBötC astrocytes reduced resting respiratory rate, minute ventilation at normoxia/eucapnia was similar to that in animals expressing control transgene (due to slight compensatory increases in tidal volume). Marked differences in ventilation between the experimental and control groups were only observed during the hypoxic challenge.

Hypercapnia: Respiratory rhythm generating circuits are silent in the absence of CO_2 and require a certain level of CO_2 to operate. Our data support the hypothesis of "distributed central chemosensitivity" which proposes that central respiratory sensitivity to CO_2 is mediated by multiple central chemoreceptor sites (one being the preBötC), with each site providing a fraction of the total response to hypercapnia and, importantly, providing tonic excitatory input in eucapnia (PMID: 10967346). The data we report in Fig 4b showing reduction in ventilation at eucapnia and hypercapnia in conditions when vesicular release mechanisms in preBötC astrocytes are blocked and hyperoxia is applied (to reduce the inputs from the peripheral chemoreceptors) – are fully consistent with this hypothesis.